# WHEN WOULD VISION-PROPRIOCEPTION POLICIES FAIL IN ROBOTIC MANIPULATION?

**Jingxian Lu**[1,2,*], **Wenke Xia**[1,2,*], **Yuxuan Wu**[3], **Zhiwu Lu**[1,2], **Di Hu**[1,2,✉]
[1]Gaoling School of Artificial Intelligence, Renmin University of China
[2]Beijing Key Laboratory of Research on Large Models and Intelligent Governance
[3]School of Artificial Intelligence, Beihang University

## ABSTRACT

Proprioceptive information is critical for precise servo control by providing real-time robotic states. Its collaboration with vision is highly expected to enhance performances of the manipulation policy in complex tasks. However, recent studies have reported *inconsistent observations* on the generalization of vision-proprioception policies. In this work, we investigate this by conducting temporally controlled experiments. We found that during task sub-phases that robot's motion transitions, which require target localization, the vision modality of the vision-proprioception policy plays a limited role. Further analysis reveals that the policy naturally gravitates toward concise proprioceptive signals that offer faster loss reduction when training, thereby dominating the optimization and suppressing the learning of the visual modality during motion-transition phases. To alleviate this, we propose the Gradient Adjustment with Phase-guidance (GAP) algorithm that adaptively modulates the optimization of proprioception, enabling dynamic collaboration within the vision-proprioception policy. Specifically, we leverage proprioception to capture robotic states and estimate the probability of each timestep in the trajectory belonging to motion-transition phases. During policy learning, we apply fine-grained adjustment that reduces the magnitude of proprioception's gradient based on estimated probabilities, leading to robust and generalizable vision-proprioception policies. The comprehensive experiments demonstrate GAP is applicable in both simulated and real-world environments, across one-arm and dual-arm setups, and compatible with both conventional and Vision-Language-Action models. We believe this work can offer valuable insights into the development of vision-proprioception policies in robotic manipulation. Videos and code are available at `https://gewu-lab.github.io/GAP/`.

## 1 INTRODUCTION

Proprioceptive information has long been recognized as a cornerstone of low-level robotic control, enabling smooth motor behavior through immediate access to the robot's internal state. This capability is especially critical in tasks requiring high accuracy and fast correction, such as posture control (Allum et al., 1998; Henze et al., 2014) and locomotion (Bjelonic et al., 2016; Lee et al., 2020; Yang et al., 2023). In recent years, there has been growing interest in introducing proprioception to learning-based manipulation (Levine et al., 2016; Cong et al., 2022; Jiang et al., 2025a). Despite the expectations that its inclusion will empower manipulation policies to maintain precision and robustness across various scenarios, existing works have reported *inconsistent observations*: HPT Wang et al. (2024) demonstrated clear improvements under the joint utilization of vision and proprioception, while Octo Octo Model Team et al. (2024) observed policies trained with additional proprioception seemed generally worse than vision-only policies. This discrepancy exposes a critical obstacle to understanding: **when vision-proprioception policies would fail in robotic manipulation?**

Extensive prior studies have revealed that the importance of visual and proprioceptive information could change over time within manipulation (Sarlegna & Sainburg, 2009; Feng et al., 2024; He et al., 2025), which referred to as *Modality Temporality*. For example, during motion-consistent phases

---

*Equal contribution, ✉ Corresponding Author (dihu@ruc.edu.cn)

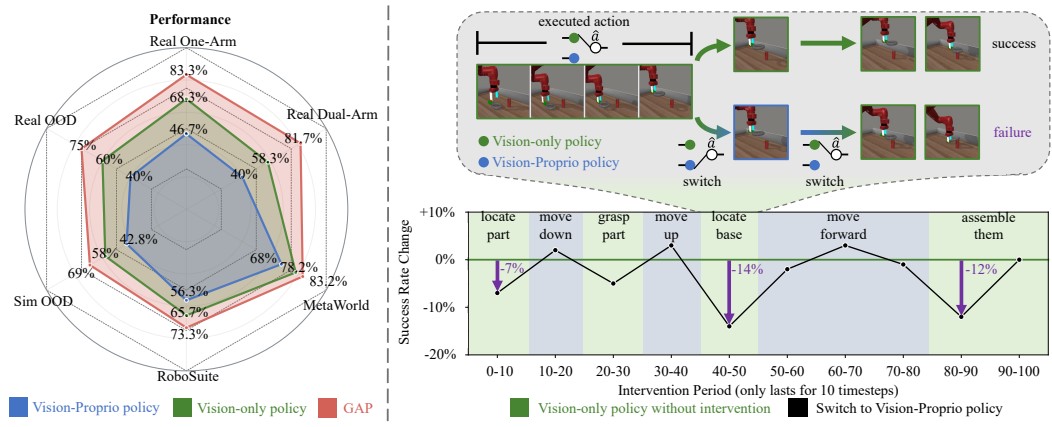

Figure 1: The generalization of vision-proprio policies. (left) Vision-Proprioception policies perform 15.8% worse than Vision-only policies. (right) We explore this through intervening the task execution of vision-only policy during different periods, by switching to vision-proprioception policy. Such intervention has minimal impact during motion-consistent phases. However, during motion-transition phases, switching leads to noticeable degradation, indicating the vision modality fails to take effect.

where the robot performs ongoing movements, the policy can benefit more from proprioceptive signals. In contrast, during the transition intervals where the robot's motion shifts, it is required to rely more on visual cues for accurate target localization. To verify whether the vision-proprioception policy exhibits such collaboration, we conduct an intervention experiment in the controlled simulation environment. Concretely, we execute the "assembly" task using the vision-only policy, but for a specific 10-timestep period, we replace executed actions with those predicted by the vision-proprioception policy under the same observations. As shown in Figure 1 (right), the intervention brings minimal impact during motion-consistent phases like "move forward", during motion-transition phases like "locate base" and "assemble them", the switching leads to noticeable degradation. It suggests that the vision modality of the vision-proprioception policy fails to take effect during motion-transition phases.

We further investigate the underlying cause from an optimization perspective. During motion-transition phases, visual cues tend to be subtle and may only differ at the pixel level (Tsagkas et al., 2025). As a result, the vision-proprioception policy naturally gravitates toward the more concise proprioceptive signals to minimize the training loss, thereby dominating the optimization (Huang et al., 2022; Fan et al., 2023). This dominance suppresses the learning of the vision modality and ultimately leads to under-utilized visual information during motion-transition phases.

To alleviate this, we propose the Gradient Adjustment with Phase-guidance (GAP) algorithm that adaptively modulates the optimization of proprioception, enabling dynamic collaboration between vision and proprioception. Specifically, we first define the motion of the robot using concise proprioception signals and segment the trajectory into motion-consistent phases. Robot's motion transits within the intervals between these phases, we thus employ a temporal network like LSTM to model transition processes. It helps eliminate potential errors introduced by the segmentation and estimates the probability that each timestep belongs to motion-transition phases. During policy learning, we guide the vision-proprioception policy to focus on essential visual cues of motion-transition phases, by applying fine-grained gradient adjustment that reduces the magnitude of proprioception's gradient.

Our GAP algorithm facilitates the vision-proprioception policy to effectively utilize proprioception without suppressing the learning of visual modality. GAP is compatible with both conventional and Vision-Language-Action models, and its versatility and effectiveness have been validated by extensive experiments in both simulated and real-world environments. Our evaluations span multiple policy architectures, including MLP-based, diffusion-based, and transformer-based policies. Further, they cover articulated object manipulation, contact-rich interactions, rotation-sensitive tasks, and soft object manipulation, across both one-arm and dual-arm robotic setups. Across all experimental settings and configurations, vision-proprioception policies equipped with our proposed GAP consistently lead to superior performances. We believe this work can offer valuable insights into the development of vision-proprioception policies in robotic manipulation.

## 2 RELATED WORK

**Vision-Proprioception Policy in Manipulation.** Vision has been the most commonly used modality in robotic manipulation policies (Zitkovich et al., 2023; Kim et al., 2024; Zeng et al., 2024). While it provides sufficient information to complete many manipulation tasks, visual data often includes a large amount of noise, such as irrelevant background distractions (Tsagkas et al., 2025). Therefore, more concise proprioceptive information has been introduced by many works to assist robotic manipulation policy, with the expectation that it can provide complementary and physically grounded information for precise and robust task execution (Cong et al., 2022; Mandlekar et al., 2022; Chi et al., 2023; Fu et al., 2024; Wang et al., 2024; Liu et al., 2024). However, existing studies have reported confused observations: some works demonstrate clear improvements when integrating proprioceptive information with vision (Cong et al., 2022; Wang et al., 2024), others observe limited gains or even detrimental effects (Mandlekar et al., 2022; Octo Model Team et al., 2024). Fu et al. (2024) attributes this to overfitting while Octo Model Team et al. (2024) suggests it arises from causal confusion between the proprioceptive information and the target actions. In this study, we further explore the utilizations of each modality and introduce a modality-temporality perspective to offer valuable insights into the development of vision-proprioception policies for robotic manipulation.

**Modality Temporality.** In manipulation tasks, each modality's contribution to decision-making can vary significantly over time. For example, in "pick-place" task, policy must first rely on vision to locate the target object. When moving toward the object, proprioception becomes more critical for executing consistent and precise actions. It is proven by strong correlations between variations in modality data and task stages (Lee et al., 2019; He et al., 2025; Jiang et al., 2025b). Feng et al. (2024) summarizes such property of manipulation tasks as modality temporality. Given this nature of robotic manipulation tasks, recent works have proposed approaches based on dynamic fusion (Li et al., 2023; Feng et al., 2024; He et al., 2025) and modality selection (Jiang et al., 2025b) to improve the performance of multimodal manipulation policies. In this study, we introduce the modality-temporality perspective to understand the roles of vision and propriocetion and propose the gradient adjustment algorithm to enhance dynamic collaboration within the vision-proprioception policy.

## 3 OPTIMIZATION ANALYSIS OF VISION-PROPRIOCEPTION POLICIES

In this section, we first formalize the problem and further analyze the generalization of vision-proprioception policies from an optimization perspective. The vision-proprioception policy is learned under the Behavior Cloning (BC) paradigm, which can be formulated as the Markov Decision Process (MDP) framework (Torabi et al., 2018). Formally, the policy $\pi$ takes the environment observation $o_t \in O$ as input at each timestep $t$. In this work, $o_t$ includes RGB-sensor readings $v_t$, and for vision-proprioception policy $\pi_{v+s}$, it includes robot proprioceptive information $s_t$ additionally. This proprioceptive information consists of the 6D pose of robot's gripper $(p_t^x, p_t^y, p_t^z, \theta_t^x, \theta_t^y, \theta_t^z) \in \mathbb{R}^6$ in Cartesian space and orientation, and a continuous value $g_t \in [0, 1]$ representing the degree of gripper opening, with $g_t = 1$ denoting fully open and $g_t = 0$ denoting fully closed.

The policy $\pi$ maps the observation history to a sequence of actions: $\hat{a}_{t+L} = \pi(o_{t-H:t})$, where $L$ and $H$ indicate the length of predicted action sequence and observation history respectively. For simplicity, we set omit them in the following discussion. The training objective can be formulated as:

$$\pi^* = \text{argmin}_\pi \mathbb{E}_{(o_t, a_t) \sim \tau_e}[\mathcal{L}_{BC}(\pi(o_t)), a_t], \tag{1}$$

where $\tau_e$ is expert demonstration dataset and $a_t$ is action labels. In vanilla BC, $\mathcal{L}$ typically represents the Mean Squared Error (MSE) loss for continuous action spaces, or Cross-Entropy (CE) loss for discrete action spaces. We focus solely on the vanilla MSE loss here.

In this work, we adopt standard joint-learning architecture to design the vision-proprioception policy, which extracts features from both vision and proprioception modalities using two separate chunks $\phi_v, \phi_s$. These features from two modalities are then concatenated and fed into the policy head $\psi$. Although some recent works have tried exploring alternative modality fusion approaches (Wang et al., 2024; Feng et al., 2024), concatenation remains the most widely used approach (Levine et al., 2016; Cong et al., 2022; Mandlekar et al., 2022). To support our analysis under this fusion approach, we split the first layer of MLP-based policy head $\psi$ into $\psi_s, \psi_v$ and rewrite the action prediction as:

$$\hat{a} = (\psi_s(f_s) + \psi_v(f_v)) \cdot W_{share} + b, \tag{2}$$

where $f_s, f_v$ is the feature extracted by $\phi_s(o), \phi_v(o)$ respectively. Under Gradient Descent (GD)-based policy learning, the optimization of the vision chunk's parameters $\omega_v$ is influenced by:

$$\frac{\partial \mathcal{L}_{BC}}{\partial \omega_v} = \frac{\partial ||\hat{a} - a||_2^2}{\partial \hat{a}} \cdot \frac{\partial (\psi_s(f_s) + \psi_v(f_v)) \cdot W_{share} + b)}{\partial f_v} \cdot \frac{\partial f_v}{\partial \omega_v}. \tag{3}$$

Within the execution trajectory of the task, changes in visual cues are usually subtle compared to proprioceptive signals. For example, when the gripper is closing, visual cues differ only at pixel-level (Tsagkas et al., 2025), while concise and low-dimension proprioceptive signals directly represent this process via changes in opening degree $g$. As a result, the vision-proprioception policy naturally gravitates toward proprioceptive signals to minimize the training loss. It leads to optimization dominated by proprioception and suppresses the learning of $\omega_v$ due to vision modality's low contribution to action prediction (Huang et al., 2022; Fan et al., 2023).

As shown in Figure 1 (right), such overreliance to proprioception brings negligible impact during motion-consistent phases, since the execution of ongoing movements benefits significantly from proprioceptive signals. However, the initial positions of the target objects vary during testing and the proprioceptive signal does not contain object-related information. During motion-transition phases, the policy is required to accurately locate the target objects. The suppressed learning of vision modality thus regretfully impairs generalization of the vision-proprioception policy*.

## 4 METHOD

To alleviate the suppression of the learning of vision modality during motion-transition phases, we propose the Gradient Adjustment with Phase-guidance (GAP) algorithm. As shown in Figure 2, we initially define the representation of robot's motion and identify motion-consistent phases. Motion-transition phase indicators are then predicted to estimate the probability that each timestep belongs to motion-transition phases. Based on these indicators, we apply fine-grained gradient adjustment during policy learning, facilitating dynamic collaboration within the vision-proprioception policy.

### 4.1 MOTION REPRESENTATION OF ROBOT

Proprioceptive signals of the trajectory $[s_1, s_2, ..., s_N]$ directly provide the state of the gripper's position $p$, orientation $\theta$, and opening degree $g$. The variations in them effectively capture the motion of the robot arm over time. We first define the representation of motion for further motion-transition phase estimation. Specifically, the motion between timestep $i$ and timestep $j$ is defined as: $m_{i:j} = \{p_{i:j}, \theta_{i:j}, g_{i:j}\}$, where $p_{i:j} = p_j - p_i$ denotes the change in the gripper's 3D position, $\theta_{i:j} = \theta_j - \theta_i$ denotes the change in orientation, and $g_{i:j} = g_j - g_i$ denotes the change in gripper opening. Together, these three dimensions provide a complete representation of the robot's motion.

### 4.2 MOTION-TRANSITION PHASE ESTIMATION

The represented motion captures the movement of robot arm, allowing expert demonstrations to be segmented into sequences of continuous states that correspond to semantically similar motions. To leverage this property for identifying motion-consistent phases, we employ the simple yet effective Change Point Detection (CPD) algorithm (Liu et al., 2013; Aminikhanghahi & Cook, 2017). The overall motion of a trajectory phase $\tau_{t_1:t_2}$ can be characterized by $m_{t_1:t_2}$. Based on whether the directions of these changes are consistent, we define the following distance between phase motion $m_{t_1:t_2}$ and adjacent motion $m_{i:i+1}$:

$$d(m_{t_1:t_2}, m_{i:i+1}) = -\cos(p_{t_1:t_2}, p_{i:i+1}) - \alpha\cos(\theta_{t_1:t_2}, \theta_{i:i+1}) - \beta(\text{sgn}(g_{t_1:t_2}) == \text{sgn}(g_{i:i+1})), \tag{4}$$

---

*For experimental results of UNet-based diffusion policies, please refer to Appendix C.3.

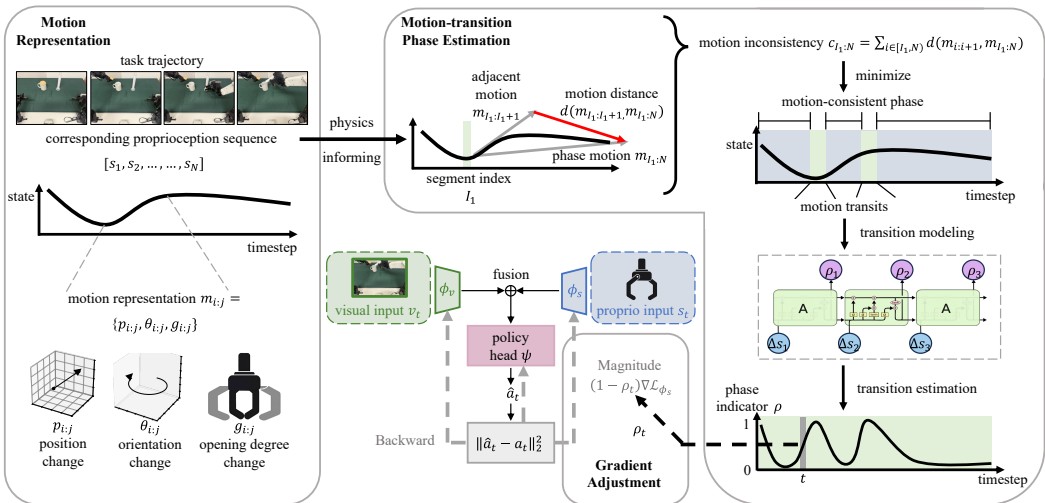

Figure 2: The pipeline of our Gradient Adjustment with Phase-guidance (GAP) algorithm. We define the motion representation and identify the motion-consistent phases by minimizing the total cost between phase motion and each adjacent motion. Motion-transition phase indicators are then estimated to reduce the magnitude of proprioception's backward gradient. GAP facilitates vision-proprioception policies to effectively utilize proprioception without suppressing vision modality.

where $\mathrm{sgn}(\cdot)$ denotes the sign function, and $\alpha, \beta$ are weighting factors for the orientation and opening degree component respectively. The statistic that measure the motion inconsistency of phase $\tau_{t_1:t_2}$ is defined as $c_{t_1:t_2} = \sum_{i=t_1}^{t_2-1} d(m_{t_1:t_2}, m_{i:i+1})$. The Change Point Detection algorithm leverages dynamic programming to identify a set of indices $I$ that minimize the total cost $\sum_I c_{\tau_I}$, segmenting the trajectory into motion-consistent phases.

Motion of the robot transits within the intervals between these phases, requiring the policy to locate target object. Vision is therefore expected to play a more significant role. However, transition is a continuous process and the discrete labels from CPD can't capture this continuous nature effectively, which is demonstrated by our ablation experiments in Section B.2 of the Appendix. Therefore, we further utilize the temporal differences of proprioceptive information $\Delta s_i = s_{i+1} - s_i$ and leverage their sequential context with an temporal network such as LSTM. It predicts motion-transition phase indicators $\rho_i$ to estimate the probability that timestep $i$ belongs to motion-transition phases. The predicted indicators $\rho$ is under the supervision of indices set $I$. Additionally, for timesteps within a range near the transition, we reduce the penalty applied to them in order to better capture the inherently continuous and gradual transition process.

## 4.3 GRADIENT ADJUSTMENT FOR MODALITY COLLABORATION

The vision-proprioception policy extracts features from both vision and proprioception modalities using two separate chunks $\phi_v, \phi_s$, which consist of an encoder and a temporal transformer, these features are then fused and fed into policy head to predict the action. However, since visual cues during motion-transition phases may be subtle, the policy tends to rely heavily on features of proprioception. As a result, the gradient optimization for corresponding samples becomes dominated by proprioceptive inputs, which in turn constrains the learning of the vision modality chunk $\phi_v$.

To mitigate this, we employ gradient adjustment to control the optimization of proprioceptive chunk $\phi_s$ during motion-transition phases, thereby guiding the vision-proprioception policy to focus more on visual cues and preventing the degradation of its generalization. Concretely, in the $j$-th epoch of Gradient Descent (GD)-based optimization, the parameters of the proprioceptive feature chunk $\omega_s^j$ are updated according to the following formula:

$$\omega_s^{j+1} = \omega_s^j - \lambda \cdot (1 - \rho) \cdot \eta \nabla \omega_s^j \mathcal{L}_{BC}(\omega_s^j), \tag{5}$$

where $\eta$ is the learning rate, $\lambda$ is a hyper-parameter that controls the degree of adjustment. For each timestep, we modulate the magnitude of the proprioception backward gradient based on its indicator $\rho$ of belonging to motion-transition phases. The higher value of $\rho$ leads to greater degree of modulation.

By applying gradient adjustment with phase-guidance as illustrated in Algorithm 1, the vision-proprioception policy is enabled to effectively leverage proprioceptive information without compromising its generalization ability.

---

**Algorithm 1** Vision-Proprioception Policy Learning with Gradient Adjustment

---

**Notations:** Expert demonstrations $o_e$, proprioceptive signals $s_e$, epoch number $T$, vision-proprioception policy $\pi_{v+s}$, proprioception chunk parameters $\omega_s$, vision chunk parameters $\omega_v$.

**Motion-Transition Phase Estimation**
Identify motion-consistent phases by Change Point Detection $I \leftarrow \text{CPD}(s_e)$;
Predict motion-transition phase indicators $\rho \leftarrow \text{LSTM}(\Delta s_e)$ ;

**Gradient Adjustment during Policy Learning**
**for** $j = 0, 1, \cdots, T-1$ **do**
    Sample a fresh mini-batch $B_j$ from expert demonstrations $o_e$;
    Feed-forward the batched data $B_j$ to $\pi_{v+s}$;
    Calculate average indicator $\rho_j$ of $B_j$;
    Update proprioception chunk $\omega_s^{j+1}$ using Equation 5;
    Update vision chunk $\omega_v^{j+1}$.
**end for**

---

## 5 EXPERIMENTS

In this section, we validate the versatility and effectiveness of our proposed Gradient Adjustment with Phase-guidance (GAP) algorithm through a series of question-driven experiments. The evaluations comprehensively cover a wide range of manipulation tasks, including articulated object manipulation, rotation-sensitive tasks, as well as long-horizon and contact-rich tasks. Ablations on each modules and hyperparameters are presented in Section B of the Appendix due to space limitations.

### 5.1 EXPERIMENTAL SETUP

We select two simulated environments as our benchmarks: Meta-World Yu et al. (2020) and Robo-Suite Zhu et al. (2020). Tasks in Meta-World are relatively simple, featuring a 4-dimensional action space that includes the gripper's position and its opening degree, while tasks in RoboSuite involve complex scenarios, longer task sequence horizons and richer physical interactions, with the action space further including the orientation of the gripper. For real-world experiments shown in Figure 3, we use a 6-DoF xArm 6 robotic arm equipped with a Robotiq gripper. Moreover, we utilize the open-source Cobot Magic platform to support tasks that require dual-arm collaboration. In all tasks, the initial position of target object varies randomly in each validation, while the initial position of gripper remains fixed. Tasks in simulation and real-world are evaluated with 100 and 20 rollouts, respectively. The detailed task descriptions are provided in Section D of the Appendix.

### 5.2 CAN GAP LEAD TO MORE ROBUST VISION-PROPRIOCEPTION POLICIES?

Vision-Proprioception policies perform generally worse than vision-only policies. Can GAP lead to more robust vision-proprioception policies? To answer this, we conducted comparative analyses between our algorithm and the following baselines:

- MS-Bot Feng et al. (2024): this method uses state tokens with stage information to guide the dynamic collaboration of modalities within multi-modality policy.

- Auxiliary Loss (Aux): following HumanPlus Fu et al. (2024), we use visual feature to predict the next frames as an auxiliary loss, which tries to enhance the vision modality.

- Mask: to prevents the overfitting to specific modality, RDT-1B Liu et al. (2024) randomly and independently masks each uni-modal input with a certain probability during encoding. We adapt the algorithm by masking only proprioception modality instead.

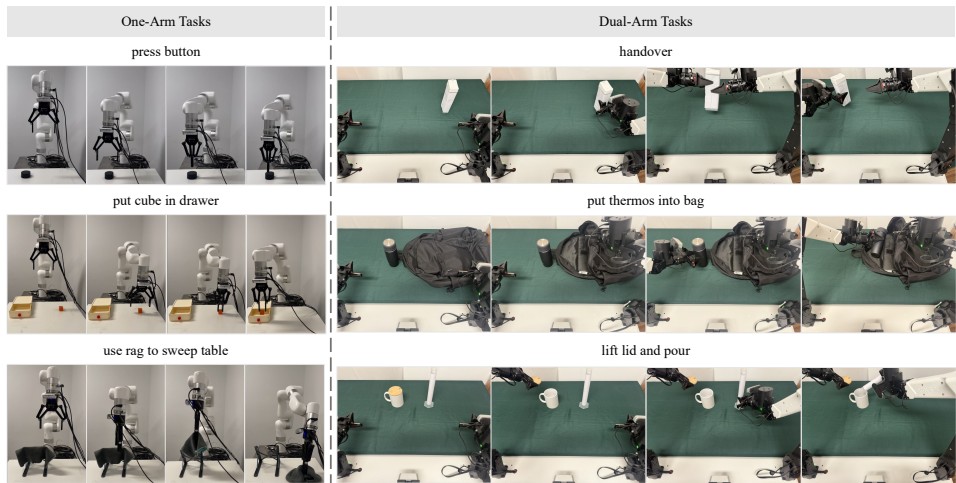

Figure 3: Visualization of real-world tasks. Our experiments cover a wide range of manipulation tasks, including both One-Arm and Dual-Arm Setups.

Results in Table 1 demonstrate that vision-proprioception policies with our GAP applied outperform vision-only policies and other methods. Although MS-Bot achieves overall improvements over the vision-only policy by incorporating stage information, it focuses on the semantic stage instead of motion-transition phases. As a result, its benefits are marginal in tasks like "push-wall" and "lift lid and pour", where motion frequently transits. This highlights the necessity of fine-grained gradient adjustment during motion-transition phases. Auxiliary loss forces the vision-proprioception policy to concentrate on visual input during the whole task, which falls short in tasks requiring proprioception to enhance the precision and robustness of manipulation, such as "threading". Meanwhile, masking the proprioceptive input with a fixed probability overlooks the modality temporality of manipulation tasks, resulting in minimal improvement. By adaptively applying fine-grained gradient adjustment during motion-transition phases, GAP enables the vision-proprioception policy to effectively leverage these two modalities and outperform both the vision-only policy and other methods.

Table 1: Comparisons with other methods in both simulated and real-world environments. Average success rate and standard deviation of simulation results are calculated over 5 seeds. The vision-proprioception policies after our gradient adjustment significantly outperform other methods.

| Suite | Meta-World | | | | | RoboSuite | | |
|---|---|---|---|---|---|---|---|---|
| Task
Method | pick-place | assembly | disassemble | push-wall | bin-picking | put hammer into drawer | stack | threading |
| Vision-only | 91.8±1.5 | 82.6±3.0 | 84.4±2.2 | 63.2±1.9 | 63.2±1.5 | 85.8±1.3 | 65.8±2.4 | 43.6±1.7 |
| Concatenation | 78.4±1.5 | 74.6±2.1 | 79.6±1.1 | 54.4±2.9 | 48.8±2.2 | 78.6±1.1 | 54.6±2.3 | 33.2±1.9 |
| MS-Bot Feng et al. (2024) | 90.8±1.6 | 91.0±2.1 | 87.8±1.9 | 66.2±1.6 | 69.4±1.9 | 87.6±1.1 | 69.4±1.9 | 51.2±1.3 |
| Aux Fu et al. (2024) | 88.2±2.6 | 90.4±1.8 | 77.8±1.9 | 51.0±3.2 | 54.4±1.5 | 71.2±1.9 | 54.0±2.2 | 45.4±2.1 |
| Mask Liu et al. (2024) | 86.4±1.8 | 89.2±1.6 | 83.8±2.7 | 79.2±3.7 | 60.4±1.1 | 78.6±1.8 | 61.8±1.9 | 47.2±1.6 |
| GAP (Ours) | 94.2±1.5 | 94.2±0.8 | 91.4±1.1 | 73.6±1.3 | 70.4±1.1 | 91.0±0.7 | 77.6±0.9 | 53.0±1.2 |

| Setup | Real One-Arm | | | Real Dual-Arm | | |
|---|---|---|---|---|---|---|
| Task
Method | press button | cube | use rag to sweep table | handover | put thermos into bag | lift lid and pour |
| Vision-only | 18/20 | 14/20 | 9/20 | 15/20 | 11/20 | 9/20 |
| Concatenation | 12/20 | 11/20 | 5/20 | 12/20 | 7/20 | 5/20 |
| MS-Bot Feng et al. (2024) | 20/20 | 16/20 | 11/20 | 16/20 | 13/20 | 10/20 |
| Aux Fu et al. (2024) | 19/20 | 16/20 | 11/20 | 15/20 | 13/20 | 8/20 |
| Mask Liu et al. (2024) | 18/20 | 14/20 | 7/20 | 15/20 | 9/20 | 7/20 |
| GAP (Ours) | 20/20 | 17/20 | 13/20 | 18/20 | 16/20 | 15/20 |

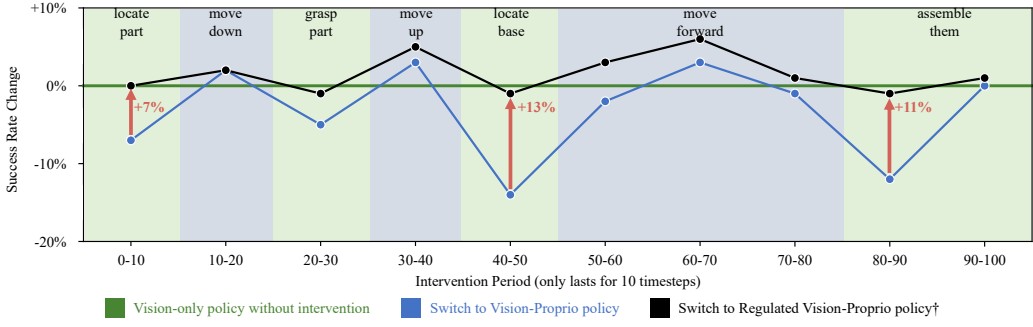

Figure 4: The intervention experiment of the GAP-equipped vision-proprioception policy. The sight changes in success rate indicate that GAP does enhance the utilization of vision modality.

## 5.3 DOES GAP ENHANCE THE UTILIZATION OF VISION MODALITY?

Although vision-proprioception policies outperform vision-only policies after apply GAP, it remains unclear whether GAP truly enhances the utilization of the vision modality within vision-proprioception policies. To answer this, we first conducted intervention experiment under the same settings as described in Section 1. As shown in the Figure 4, the degrees of suppression of vision modality during motion-transition phases are significantly reduced after applying GAP, indicating GAP does enhance the utilization of vision modality. We further evaluated the generalization of the vision-proprioception policies in out-of-distribution (OOD) scenarios. In each scenario, the initial distribution of object positions differs from that in the training dataset of expert demonstrations. The vision-only policies are less affected by such changes due to well-utilized vision modality as demonstrated in Tabel 2. Vision-proprioception policies exhibit poor generalization with suppressed vision. Meanwhile, Our algorithm alleviates this by regulating the optimization of the proprioceptive, preventing the suppression. The maintained superior performance over vision-only policy also indicates the effectiveness of introducing proprioception modality for precise and robust manipulation. The linear-probing experiments that directly evaluate the vision modality are provided in Section C.1.

Table 2: Experiments under out-of-distribution settings. For each task, our proposed GAP algorithm enhances the generalization of the vision-proprioception policies.

| Setup | Meta-World | | RoboSuite | | Real One-Arm | Real Dual-Arm |
|---|---|---|---|---|---|---|
| Method \ Task | assembly | bin-picking | stack | threading | cube | handover |
| Vision-only | 78% | 59% | 63% | 32% | 12/20 | 12/20 |
| Concatenation | 62% | 32% | 49% | 28% | 7/20 | 9/20 |
| GAP (Ours) | 88% | 67% | 72% | 49% | 15/20 | 15/20 |

## 5.4 IS GAP COMPATIBLE WITH VISION-LANGUAGE-ACTION MODELS?

Above experiments have demonstrated that our algorithm facilitates dynamic collaboration within conventional vision-proprioception models. We further investigate is GAP compatible with Vision-Language-Action (VLA) models. Specifically, we compare fine-tuned Octo model Octo Model Team et al. (2024) using only visual information (Octo-V) versus using both vision and proprioception (Octo-VP), and tries to apply our gradient adjustment algorithm during fine-tuning. As reported in the original paper, policies trained with additional propioception seemed generally worse than vision-only policies. We observe the same trend across various tasks in Table 3. However, after applying our gradient adjustment algorithm, Octo-VP† achieves an average improvement of 17%

and exhibits stronger generalization ability than Octo-V. These results suggest that our algorithm effectively enhances dynamic collaboration between vision and proprioception within VLA models.

Table 3: Performances of fine-tuned Octo. † indicates GAP is applied.

| Suite | Meta-World | | RoboSuite | |
|---|---|---|---|---|
| Model ╲ Task | disassemble | push-wall | put hammer into drawer | threading |
| Octo-V | 95% | 77% | 92% | 69% |
| Octo-VP | 82% | 65% | 88% | 57% |
| Octo-VP† | 100% | 85% | 97% | 78% |

## 5.5 CAN GAP BE APPLIED TO VARIOUS MODALITY FUSION APPROACHES?

The preliminary results in Section 3 reveal that the vision-proprioception policy using straightforward concatenation tends to perform worse than the vision-only policy. We further explore a broader set of fusion approaches and validate the versatility of our algorithm. Specifically, we apply GAP to three typical and widely used fusion approaches: Concatenation, Summation and FiLM Perez et al. (2018).

As reported in Table 4, vision-only policies outperforms all three fusion approaches in tasks such as "pick-place" and "put hammer into drawer", indicating that vision modality suffice for certain tasks. However, they fail drastically in task "threading" due to demands for precise manipulation and exhibits suboptimal performance in task "push-wall", which involves visual occlusions at the target location, highlighting the necessity of the inclusion of proprioceptive information for precise and robust manipulation.

Table 4: Performance of typical fusion approaches combined with GAP. † indicates GAP is applied.

| Suite | Meta-World | | | | | RoboSuite | | |
|---|---|---|---|---|---|---|---|---|
| Method ╲ Task | pick-place | assembly | disassemble | push-wall | bin-picking | put hammer into drawer | stack | threading |
| Vision-only | 92% | 82% | 85% | 64% | 63% | 86% | 67% | 44% |
| Concatenation | 79% | 76% | 80% | 56% | 49% | 79% | 56% | 34% |
| Summation | 78% | 95% | 80% | 54% | 61% | 75% | 49% | 30% |
| FiLM | 75% | 91% | 47% | 67% | 59% | 76% | 53% | 41% |
| Concatenation† | 94% | 96% | 91% | 73% | 70% | 91% | 77% | 52% |
| Summation† | 92% | 97% | 93% | 66% | 70% | 88% | 82% | 48% |
| FiLM† | 90% | 94% | 85% | 74% | 68% | 95% | 72% | 46% |

Concatenation preserves raw features from both modalities, but the high-dimensional redundancy hinders the policy to dynamically utilize each modality. As a result, it underperforms in tasks like "push-wall", where effective coordination is required. Simple summation may obscure critical details, whose limitation is evident in precise manipulation tasks such as "threading" and "push-wall". Meanwhile, FiLM applies affine transformations to conditionally adjust features, making it more suitable for tasks requiring modality collaboration. For instance, it achieves a notably higher score in "push-wall" task. However, its performance tends to degrade in simpler tasks where such complex conditioning may be unnecessary. Conversely, GAP successfully unlocked the full potential of the vision-proprioception policy, outperforming vision-only policies in all three fusion approaches.

## 6 VISUALIZATION

### 6.1 VISUALIZATION OF MOTION-TRANSITION PHASE ESTIMATION

To better understand the Motion-Transition Phase Estimation, we present visualizations that display the estimated transition probabilities $\rho$ alongside corresponding RGB images of "threading" task in this section. We also show the relationship between $\rho$ and visual uncertainty. Specifically, we leverage R3M (Nair et al., 2023) to extract universal visual features of this task, and calculate the

local entropy of features based on a sliding window covariance, denoted as $u$. The visual uncertainty $u$ is then normalized to the range $[0, 1]$ using $(\tanh(u) + 1)/2$. As shown in Figure 5, the value of $\rho$ increases sharply at key moments, such as when the robotic arm grasps an object or transitions from a vertical to a rotational movement. This demonstrates that our method accurately captures motion transitions during the robot's operation. Moreover, we observe that increases in $\rho$ are consistently accompanied by decreases in scaled visual uncertainty, indicating a clear inverse relationship between them. It implies that changes in the visual input tend to be relatively subtle during motion transitions, highlighting the importance of enhancing the learning of the visual modality.

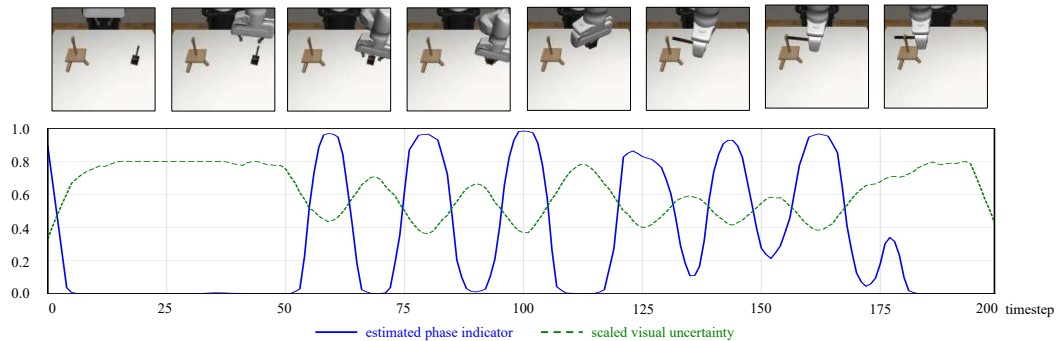

Figure 5: Visualization of Motion-Transition Phase Estimation.

## 6.2 VISUALIZATION OF LOSS CURVES

Furthermore, we demonstrate the effect of GAP on the loss curves of vision-proprioception policy in Figure 6. Due to the adjustment of GAP to the gradient updates for proprioceptive parameters, the loss decreases more slowly during the early and middle stages of training. However, the policy ultimately converges to a lower loss, indicating that GAP effectively enhances the performance of vision-proprioception policies.

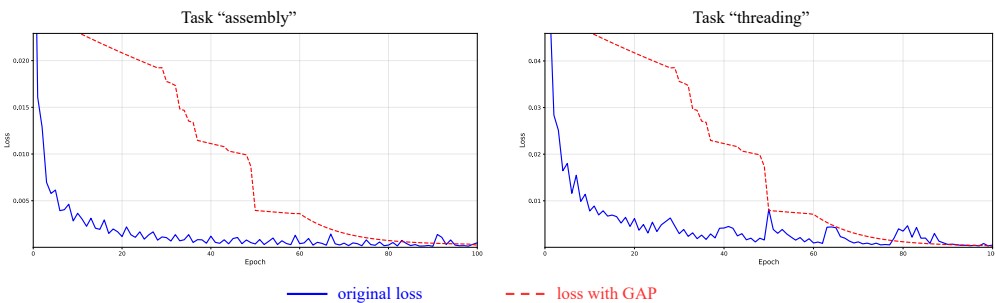

Figure 6: Visualization of loss curves.

## 7 CONCLUSION AND LIMITATION

In this work, we illustrate that the vision modality of the vision-proprioception policy plays a limited role during motion-transition phases due to suppression. To alleviate this, we propose the Gradient Adjustment with Phase-guidance (GAP) algorithm, enabling dynamic collaboration between vision and proprioception within vision-proprioception policy. We believe this work can offer valuable insights into the development of vision-proprioception policies for robotic manipulation.

**Limitations.** All vision-proprioception policies are trained on single embodiment in this work. As existing large-scale datasets often contain diverse embodiments, exploring the role of proprioception in cross-embodiment datasets would be promising for future research.

ACKNOWLEDGMENTS

This work was supported by Beijing Natural Science Foundation (4262050), in part by National Natural Science Foundation of China (62376274) and the fund for building world-class universities (disciplines) of Renmin University of China.

REPRODUCIBILITY STATEMENT

Extensive efforts have been made to ensure the reproducibility of our work. Descriptions of datasets and experiment settings are provided in Section 5.1 and Appendix A.1. Hyperparameters are presented in Appendix A.2. Furthermore, the algorithmic procedure is clearly described in Algorithm 1.

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

APPENDIX

# A  IMPLEMENTATION DETAILS

In this section, we provide implementation details, including experimental settings and algorithmic specifics, to facilitate the reproduction of our work.

## A.1  EXPERIMENT SETTINGS

Simulation experiments are conducted on two representative environments: Meta-World Yu et al. (2020) and RoboSuite Zhu et al. (2020). Tasks in Meta-World are relatively simple, featuring a 4-dimensional action space that includes the gripper's position and its opening degree, while RoboSuite tasks involve complex scenarios, longer task sequence horizons and richer physical interactions, with the action space further including the orientation of the gripper. During policy learning, task in Meta-World uses 100 hard-coded expert demonstrations from its official open-source implementation, while task in RoboSuite leverages 500 synthesized trajectories that generated from human demonstrations (Mandlekar et al., 2023).

We use an observation history window $H$ of 5 with the learning rate $3e^{-4}$ to train the vision-proprioception policy network. The parameters of the policy are optimized by Adam. The vision-proprioception policy consists of two feature extraction chunks and a policy head. For the visual input, spatial features are first extracted using a ResNet-18 He et al. (2016) backbone, then projected into a 512-dimensional hidden representation. A 4-layer temporal transformer with a hidden dimension of 256 is applied to capture temporal features. For proprioceptive input, spatial features are extracted using a 3-layer MLP. These features are fused via concatenation and then passed through a 3-layer MLP to output a sequence of actions of length $L = 9$. We train with batch size 128 on a single NVIDIA RTX 3090 GPU for 100 epochs (around 2 hours for 100 Meta-World demonstrations and 8 hours for 500 RoboSuite demonstrations).

For real-world experiments, we use a 6-DoF xArm 6 robotic arm equipped with a Robotiq gripper for all one-arm tasks. The visual information is provided by RGB images captured by an Intel RealSense D435i camera mounted on the wrist. Moreover, we utilize the open-source Cobot Magic platform to support tasks requiring dual-arm collaboration. It features four robot arms and three Intel RealSense D435 RGB-D cameras. Due to extended reach of the dual-arm setup, the two wrist-mounted cameras can't fully capture the task scene. Therefore, we also incorporate the front camera on the platform to obtain a more comprehensive RGB observation of the task. Policies of real-world experiments are trained with 50 demonstrations collected by teleoperation. For one-arm tasks, the vision-proprioception policy is the same as in simulation. For dual-arm tasks, we modified the ACT Zhao et al. (2023) architecture. we train the left-arm policy using observations from the left-wrist camera, the front camera, and the left-arm proprioceptive information, and similar settings are applied to the right-arm policy.

In all tasks, the initial position of target object varies randomly in each validation, while the initial position of gripper remains fixed. In out-of-distribution (OOD) scenarios, the initial distribution of object positions differs from that in the training dataset of expert demonstrations.

## A.2  DETAILS OF GAP

In this study, we illustrate that vision-proprioception policy would fail during motion-transition phases due to its suppressed vision modality. To alleviate this, we propose the Gradient Adjustment with Phase-guidance (GAP) algorithm, enabling dynamic collaboration between vision and proprioception within vision-proprioception policy.

The complete list of hyperparameters used in Equations 4 and 5 is provided in Table 5. Since the motion inconsistency of the gripper's opening state is measured using binary (0/1) values, we set $\beta$ to a lower value to balance the numerical scale, while keeping $\alpha = 1$ to account for the overall gripper motion. To enable fine-grained gradient adjustment, we set the parameter $\lambda$ controlling the degree of adjustment to a moderate level, avoiding excessive modulation leading to unstable gradient updates and potential policy collapse. Additionally, we apply gradient adjustment only during the early stage of policy learning, such as the first 50 epochs. Ablation studies on these hyperparameters

are provided in Section B.3. For bimanual tasks, the proprioceptive information from each arm differs in physical meaning. The trajectories of the two arms thus have distinct motion transition phases. Therefore, we train policies for each arm independently. In transformer-based models, modalities become highly integrated within the transformer, we thus only applied gradient adjustment to the proprioception feature extractor (i.e., the parameters before the transformer) for Octo-VP.

Table 5: List of hyperparameters uesd in GAP.

| Hyperparameters | Value |
|:---:|:---:|
| $\alpha$ | 1 |
| $\beta$ | $2e^{-3}$ |
| $\lambda$ | 0.3 |

## B    ABLATION EXPERIMENTS

### B.1    ABLATION ON MOTION-TRANSITION PHASE ESTIMATION

In our proposed GAP framework, the Motion-Transition Phase Estimation module is essential for enhancing vision-proprioception policies. To thoroughly analyze its effectiveness and robustness, we conducted comprehensive ablation studies.

First, we kept other components unchanged and tested the impact of different trajectory decomposition methods on policy learning. The detailed descriptions of the comparison methods are as follows:

- Human: Since obtaining ground truth phase transition labels is infeasible due to the complexity and diversity of manipulation tasks, we instead employ human experts to manually label phase transitions.

- HDBSCAN: Following Emma-X Sun et al. (2024) , we utilize the HDBSCAN algorithm to cluster our distance metric defined in Equation 4, decomposing trajectories into phases.

- CoTPC Jia et al. (2024): To decompose task trajectories into temporally close and functionally similar subskills, CoTPC applies the Change Point Detection (CPD) algorithm on the sequence of robot actions using Cosine distance.

As shown in Table 6, using human-labeled phase transitions for gradient adjustment does provide a slight improvement in performance. However, since human annotators tend to decompose trajectories from a semantic perspective rather than based on actual motion changes, this approach may miss certain transition points, resulting in only limited gains. The cluster-based HDBSCAN method disrupts the temporal structure inherent to manipulation tasks, leading to even worse policy performance compared to human labeling. Notably, both CoTPC and our GAP framework employ the CPD algorithm for phase estimation and generally outperform human annotation across most tasks, suggesting that automated phase estimation can capture motion transitions more effectively than manual labeling. However, CoTPC relies on a simple cosine distance, which lacks the ability to represent the nuanced characteristics of robot motion, thus yielding less significant performance improvements compared to GAP. In our approach, the motion-consistent distance defined in Equation 4 captures spatial position changes, rotational changes, and gripper opening state changes, enabling more accurate and efficient prediction of motion transitions.

To further understand the performance gap between GAP and human labeling, we calculated the precision and recall between GAP outputs and human labels. Our results show that GAP achieves a high recall (97% on average) but a relatively lower precision (73% on average), suggesting that the CPD algorithm not only accurately detects semantic changes during manipulation but also identifies motion transitions that may be overlooked by humans. This experiment highlights that our method offers an effective and efficient approach, given that obtaining ground truth phase transition labels is infeasible due to the complexity and diversity of manipulation tasks.

Table 6: Performance Comparison between GAP and other decomposition methods.

| Suite | Meta-World | | | | | RoboSuite | | |
|---|---|---|---|---|---|---|---|---|
| Method / Task | pick-place | assembly | disassemble | push-wall | bin-picking | put hammer into drawer | stack | threading |
| Vision-only | 92% | 82% | 85% | 64% | 63% | 86% | 67% | 44% |
| Human | 95% | 83% | 88% | 68% | 65% | 88% | 70% | 47% |
| HDBSCAN Sun et al. (2024) | 88% | 81% | 87% | 60% | 57% | 82% | 61% | 37% |
| CoTPC Jia et al. (2024) | 91% | 85% | 90% | 66% | 62% | 85% | 69% | 50% |
| GAP (Ours) | 94% | 96% | 91% | 73% | 70% | 91% | 77% | 52% |

## B.2 ABLATION ON TEMPORAL NETWORK

We further validate the effectiveness of the LSTM component by conducting comprehensive ablation studies. Specifically, we compared the LSTM-based phase transition modeling with the following alternatives:

- Fixed $\rho$: For the phase transition indices output by CPD, we applied a fixed gradient adjustment magnitude $\rho$ to the corresponding samples during policy learning.

- Smooth: Following KOI Lu et al. (2025), we modeled the transition process by treating the CPD output index as the mean of a Gaussian distribution.

The results in Table 7 show that while alternative methods such as smoothing may serve as substitutes for modeling transitions, they regretfully fail to provide significant performance improvements.

Table 7: Ablation Studies on the LSTM component.

| Method / Task | put hammer into drawer | threading | cube |
|---|---|---|---|
| Vision-only | 86% | 44% | 70% |
| Fixed $\rho$=0.3 | 77% | 44% | 65% |
| Fixed $\rho$=0.5 | 89% | 48% | 75% |
| Fixed $\rho$=0.7 | 82% | 38% | 65% |
| Smooth | 88% | 46% | 80% |
| GAP (Ours) | 91% | 52% | 85% |

Additionally, to demonstrate the LSTM's effectiveness in eliminating potential estimation errors, we conducted noise injection experiments by randomly selecting a direction and shifting each CPD output index with a probability of 0.5, repeating this shift process recursively. As shown in Table 8, even when considerable noise is injected into the CPD outputs, the policy performance remains robust. We attribute this robustness to the LSTM's capability to model continuous transitions, which effectively alleviates the impact of errors introduced by the CPD algorithm.

Table 8: Noise Injection Experiments.

| Method / Task | put hammer into drawer | threading | cube |
|---|---|---|---|
| Vision-only | 86% | 44% | 70% |
| Concatenation | 79% | 34% | 55% |
| Noise-Injected | 88% | 49% | 85% |
| GAP (Ours) | 91% | 52% | 85% |

Table 9: Ablation Studies on $\alpha$ and $\beta$.

| Method \ Task | assembly | threading | put the banana in the box |
|---|---|---|---|
| Vision-only | 83% | 44% | 55% |
| Concatenation | 75% | 33% | 50% |
| $\alpha = 0.5, \beta = 2e^{-3}$ | 93% | 49% | 65% |
| $\alpha = 2, \beta = 2e^{-3}$ | 91% | 57% | 60% |
| $\alpha = 1, \beta = 10e^{-3}$ | 88% | 51% | 60% |
| GAP ($\alpha = 1, \beta = 2 \times 10^{-3}$) | 94% | 53% | 65% |

### B.3 ABLATIONS ON HYPERPARAMETERS

To validate the effectiveness and robustness of our proposed GAP framework, we conduct comprehensive ablation studies on the key hyperparameters of the gradient adjustment.

First, we examine the selection of hyperparameters $\alpha$ and $\beta$, which serve to balance the relative contribution of each term in Equation 4, thus providing a more accurate measurement of whether the directions of these changes are consistent. As described in Section A.1, we set $\alpha = 1$ so that displacement consistency and rotation consistency are equally weighted. Since gripper opening consistency is measured via sign function (which outputs 0 or 1), we set $\beta = 2e^{-3}$ to ensure this term would not overshadow the first two terms. The results in Table 9 show that when $\alpha$ is small, displacement consistency contributes more to overall motion consistency, which leads to higher success rate of tasks where displacement dominates, such as "assembly" and "put the banana in the box". In contrast, when $\alpha = 2$, the final policy performs better on the "threading" task, where rotation is more important. Increasing $\beta$ to $10 \times 10^{-3}$ to emphasize the importance of gripper opening does not yield substantial improvement. Notably, all three sets of hyperparameter choices result in vision-proprioception policies that outperform policies without GAP, and are also superior to vision-only policies, indicating that our proposed GAP method is robust to hyperparameter selection.

We also perform ablation studies on the $\lambda$ in Equation 5. In our GAP framework, we set $\lambda = 0.3$ by default to control the degree of gradient adjustment. As shown in Table 10, the policy performance is not highly sensitive to changes in $\lambda$ between 0.2 and 0.4. However, setting $\lambda$ too small results in insufficient attenuation of the proprioception gradients, causing the policy performance to become similar to or even worse than the simple concatenation baseline. Conversely, if $\lambda$ is set too large, it leads to instability and even collapse during policy training.

Table 10: Ablation Studies on $\lambda$.

| Method \ Task | assembly | threading | put the banana in the box |
|---|---|---|---|
| Vision-only | 83% | 44% | 55% |
| Concatenation | 75% | 33% | 50% |
| $\lambda = 0.1$ | 77% | 41% | 50% |
| $\lambda = 0.2$ | 91% | 57% | 60% |
| $\lambda = 0.4$ | 96% | 49% | 65% |
| $\lambda = 0.8$ | 70% | 37% | 50% |
| GAP ($\lambda = 0.3$) | 94% | 53% | 65% |

The impact of stages of applying GAP has been evaluated as well. In all of our experiments, policies are trained for 100 epochs. We apply GAP during the first 50 epochs to avoid excessive modulation, which may cause unstable gradient updates and potentially lead to policy collapse. Table 11 presents the impact of the number of training stages $x$ during which GAP is applied on the final policy

performance. Applying GAP in the early stages of training improves the performance of vision-proprioception policies. However, if the gradient updates for proprioception are suppressed for too many epochs, the performance of policies deteriorates.

Table 11: Ablation Studies on the number of stages of applying GAP $x$.

| Method / Task | assembly | threading | put the banana in the box |
|---|---|---|---|
| Vision-only | 83% | 44% | 55% |
| Concatenation | 75% | 33% | 50% |
| $x = 10$ | 77% | 38% | 50% |
| $x = 30$ | 88% | 47% | 60% |
| $x = 70$ | 83% | 45% | 55% |
| $x = 90$ | 72% | 35% | 50% |
| GAP ($x = 50$) | 94% | 53% | 65% |

## C  ADDITIONAL EXPERIMENTS

### C.1  LINEAR-PROBING EXPERIMENTS ON THE VISUAL FEATURES

To directly evaluate the visual features and verify whether our proposed GAP framework enhances the learning of visual representations, we conduct linear-probing experiments on the vision features extracted before and after applying GAP. Specifically, we extract and freeze the visual branch of the vision-proprioception policy and train a separate policy head to predict actions using only these frozen visual features. This linear-probing setup ensures that any performance differences directly reflect the quality of the learned visual representations.

These policies are evaluated in both in-distribution and out-of-distribution settings, where object positions at test time differ from those in the training data. As shown in Table 12, the visual branch trained with GAP significantly outperforms that without GAP in both in-distribution and out-of-distribution scenarios across different tasks.

Table 12: Linear-Probing Experiments on the Visual Features.

| Method / Task | assembly | assembly (OOD) | threading | threading (OOD) |
|---|---|---|---|---|
| Visual branch without GAP | 61% | 45% | 26% | 14% |
| Visual branch with GAP | 74% | 69% | 41% | 27% |

### C.2  MORE OUT-OF-DISTRIBUTION (OOD) EVALUATIONS

To further assess the robustness of GAP under diverse visual distribution shifts, we conducted additional out-of-distribution experiments beyond the object placement variations reported in Table 2. Specifically, we evaluated three additional types of visual perturbations:

- Table Color: The color of the table was modified during testing, altering the overall scene appearance.

- Object Color: Objects with different colors from those used during training were employed, introducing color-based perturbations.

- Lighting: Colored spot lighting was introduced to create additional visual disturbances during testing.

The results in Table 13 and Table 14 show that GAP consistently outperforms baseline methods under all tested OOD conditions. Among different perturbations, placement changes exert the most significant impact on performance, underscoring the importance of accurate target object localization for successful manipulation. While concatenation benefits from proprioceptive information that remains unaffected by visual disturbances, the lack of precise position perception results in overall suboptimal performance.

In contrast, GAP achieves superior performance compared to vision-only approaches, highlighting that leveraging proprioceptive information in collaboration with vision enhances policy robustness under challenging OOD scenarios.

Table 13: OOD evaluation of the task "assembly" in Meta-World.

| Task
Method | Placement | Table Color | Object Color | Lighting |
|---|---|---|---|---|
| Vision-only | 78% | 80% | 75% | 73% |
| Concatenation | 62% | 72% | 76% | 74% |
| GAP (Ours) | 88% | 85% | 83% | 90% |

Table 14: OOD evaluation of the task "cube" in the Real-World.

| Task
Method | Placement | Table Color | Object Color | Lighting |
|---|---|---|---|---|
| Vision-only | 60% | 65% | 60% | 55% |
| Concatenation | 35% | 50% | 55% | 50% |
| GAP (Ours) | 75% | 80% | 85% | 80% |

### C.3  EXPERIMENTS WITH VARIOUS POLICY HEADS

The previous experiments have demonstrated the effectiveness of our GAP on policies with MLP-based policy heads. Additionally, we conducted experiments using the popular UNet-based diffusion policy head Chi et al. (2023) and reported the results in Table 15. The results show that policies using a diffusion-based policy head are generally better those with MLP-based heads, while applying the GAP method still outperforms vision-only policies and other methods.

Table 15: Comparisons with other methods in simulated environments. Vision-Proprioception policies are equipped with UNet-based diffusion policy head.

| Suite | Meta-World | | | RoboSuite | |
|---|---|---|---|---|---|
| Task
Method | disassemble | push-wall | bin-picking | stack | threading |
| Vision-only | 92% | 70% | 66% | 73% | 51% |
| Concatenation | 82% | 62% | 51% | 62% | 40% |
| MS-Bot Feng et al. (2024) | 93% | 75% | 72% | 77% | 58% |
| Aux Fu et al. (2024) | 93% | 71% | 75% | 72% | 55% |
| Mask Liu et al. (2024) | 87% | 83% | 67% | 66% | 53% |
| GAP (Ours) | 97% | 88% | 81% | 87% | 63% |

### C.4  EXPERIMENTS OF EXTENDING GAP TO ON-POLICY UPDATES

To verify whether GAP can be effectively applied to on-policy updates, we conduct experiments using online learning settings in this section. In these experiments, we use vision-proprioception policies initialized with behavior cloning and train the LSTM to predict motion transition probabilities in

real-time during exploration, thereby adjusting proprioception gradient updates dynamically in online learning. This setup allows us to assess whether GAP maintains its effectiveness when the policy is updated based on newly collected experience, rather than a fixed offline dataset.

Table 16: Experiments with On-Policy Updates.

| Method \ Task | assembly | bin-picking | threading |
|---|---|---|---|
| Vision-only | 88% | 71% | 50% |
| Concatenation | 78% | 50% | 37% |
| GAP | 94% | 82% | 61% |

The results in Table 16 demonstrate that policies trained with online learning are more robust, and GAP can be effectively applied to on-policy updates across different tasks and environments. Additionally, the online learning experiments demonstrate that GAP does not over-dampen necessary proprioception reliance, as the policies maintain strong performance while still benefiting from the enhanced visual representations.

## D    TASK DESCRIPTIONS

In this subsection, we provide detailed descriptions of tasks we evaluated and visualize additionally simulation tasks in Figure 7. The evaluations comprehensively cover a wide range of manipulation tasks, including simple pick-and-place tasks, rotation-sensitive tasks, as well as long-horizon and contact-rich tasks and includes one-arm and dual-arm robot setups.

- pick-place: The robot arm first needs to locate the position of the cylinder, then move toward it. After grasping the object, it locates the target circle and moves the object to the target position.

- assembly: The robot arm first locates the position of the part, then moves toward it. After grasping the part, it locates the position of the base, moves the part above it, and finally assembles them.

- disassemble: The robot arm first locates the position of the part, then moves toward it. After grasping the part, it disassembles the part from the base.

- push-wall: The robot arm first locates the position of the cylinder, then moves toward it. After grasping it, the arm pushes it to a target point behind the wall. The target point is not visible through visual observation, but its location remains fixed across all evaluations.

- bin-picking: The robot arm first locates the position of the green cube, then moves toward it. After grasping it, the arm removes it from the red container and finally places it into the blue container.

- put hammer into drawer: The robot arm first locates the position of the drawer and then moves toward it. After opening the drawer, it locates the position of the hammer, grasps it, places it inside the drawer, and finally closes the drawer.

- stack: The robot arm first locates the position of the red cube and then moves toward it. After grasping it, it locates the position of the green cube and finally stacks the red cube on top of the green cube.

- threading: The robot arm first locates the position of the pin and then moves toward it. After grasping it, it locates the position of the pinhole, then moves and rotates to align the pin with the hole, and finally inserts the pin into the hole.

- press button: The robot arm first locates the position of the button, then moves toward it, closes the gripper, and presses the button.

- cube: The robot arm first locates the position of the cube, then moves toward it. After grasping the cube, it locates the drawer and finally places the cube in the drawer.

- use rag to sweep table: The robot arm first locates the position of the rag, then moves toward the rag. It rotates the arm to grasp the rag, takes it down from the shelf, and finally rotates the rag to sweep the table.

- handover: The robot first locates the position of the rectangular block, then its right arm moves toward it. After grasping the block, the right arm moves it to the center, where the left arm takes over the block in a handover. Finally, the left arm places it on the left side of the table.

- put thermos into bag: The robot first locates the position of the bag, then its right arm moves toward it, lifts it, and rotates the wrist to open the bag. Next, robot locates the thermos, uses the left arm to grasp it, and puts it into the bag.

- lift lid and pour: The robot first locates the position of the cup, then its left arm moves toward it to grasp and lift the lid. Next, robot locates the graduated cylinder, uses the right arm to grasp it, moves it above the cup, and finally rotates the cylinder to pour.

## E    THE USE OF LARGE LANGUAGE MODELS (LLMS)

To improve the clarity and quality, this work uses LLMs for refining grammar.

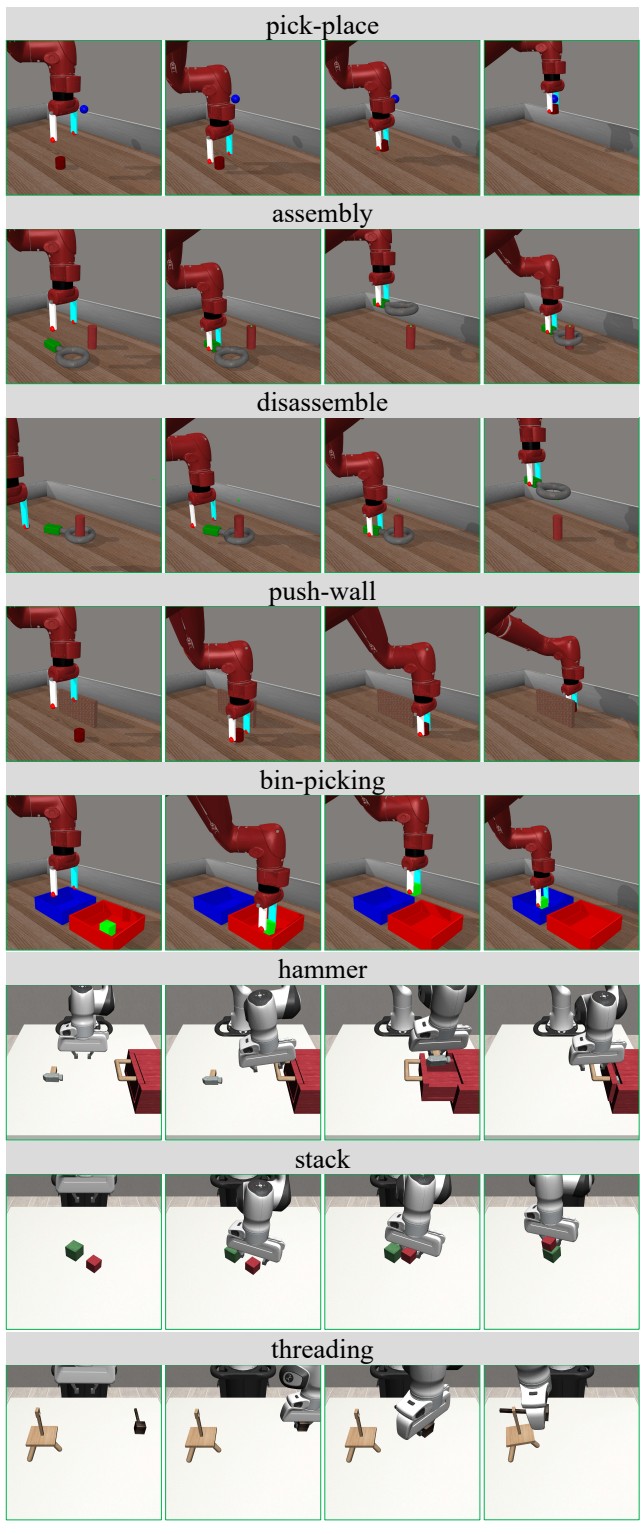

Figure 7: Visualization of tasks we evaluated in simulation environments.

