# OpenReview forum: "When would Vision-Proprioception Policies Fail in Robotic Manipulation?"
_ICLR.cc/2026/Conference — ICLR 2026 Poster_

### Official Review · Reviewer_rSWj · 2025-10-23

**Soundness:** 3
**Presentation:** 3
**Contribution:** 2
**Rating:** 4
**Confidence:** 5

**Summary:**

The paper claims that in vision-proprioception policies, the proprioceptive modality, which is relatively easier to learn, suppresses the visual modality during the motion-transition phases of a task. The authors propose a gradient adjustment algorithm that effectively addresses this issue, supported by well-designed experiments in both real and simulated environments. In particular, the proposed method first performs an initial segmentation of the task's motion-transition phases using a Change Point Detection (CPD) algorithm on proprioceptive data. It then uses a temporal model, such as an LSTM, to refine this and calculate the probability of each time step belonging to a motion-transition phase. Based on this probability, it then applies dynamic gradient adjustment to regulate learning from proprioception, thereby preventing the suppression of the visual modality. The authors validate the effectiveness of this method across various settings, including simulation, the real world, and with different architectures like conventional policies and Vision-Language-Action models.

**Strengths:**

1. This paper provides a crucial diagnosis for the inconsistent performance of vision-proprioception policies, a known but previously under-explained problem.
2. The paper's argument is well-structured and easy to follow, building a logical methodology directly from its clear problem diagnosis.
3. The paper validates the effectiveness of the proposed methodology by demonstrating high task success rates throughout its extensive experiments and ablation studies.

**Weaknesses:**

1. Reliance on Proprioceptive Cues: The reliance on proprioceptive cues for phase detection could be a failure point in tasks where transitions are defined by external, non-physical events (e.g., waiting for a visual cue), as the algorithm may not detect these critical moments.
2. Lack of Direct Evidence for Phase Estimation: The paper's central mechanism, i.e., the phase estimation, is not empirically validated with visualizations. The absence of plots showing the phase changes along with RGB images and/or the estimated transition probabilities makes it difficult to verify that this core component is functioning as intended, independently of the final task outcomes.  The final success rates are not enough to validate the rationale of this proposed method.

**Questions:**

1. Please include comparison plots (in the supplement) of the loss curves with and without your method for a given task to clearly demonstrate the effectiveness of the proposed approach.
2. Please add some plots to show how the phase changes  (the estimated transition probabilities) along with scene RGB images for validation.
3. In Figure 4, the regulated policy still shows slightly lower performance than the vision-only policy during some motion transition stages. Could you analyze the reason for this?
4. In Table 4, for the “disassemble” task, the success rate of the FiLM fusion method is significantly lower than other methods. What might be the reason for this specific underperformance?
5. In the real-world bimanual task experiment, why were the policies for the left and right arms trained separately instead of as a single, unified policy?
6. The paper states that for training stability, gradient adjustment is only applied during the early stages of policy learning. Could you provide a more precise analysis of this reasoning, ideally supported by evidence such as training-related plots?
7. To further enhance the paper’s replicability and transparency, the reviewer recommends submitting the source code as a part of the supplementary material.

---

> ### Author Response · Authors · 2025-11-23
>
> We acknowledge the reviewer's valuable feedback and appreciate the opportunity to respond to each of concerns below.
>
>
> $\color{blue}Question \space 1: $ Reliance on proprioceptive cues, and the phase estimation may fail in scenarios where transitions are defined by external, non-physical events.
>
> $\color{darkgreen}Response \space 1: $
> We thank the reviewer for raising this important question. In expert demonstrations, transitions driven by visual cues are reflected in changes in the expert’s actions. For example, the expert policy only begins to act when a specific visual cue appears, and these transitions are subsequently recorded in the proprioception of the expert trajectory. Because GAP first post-processes the expert trajectories as described in Algorithm 1, it is able to employ CPD and LSTM to effectively analyze the recorded proprioceptive signals and reliably detect motion transition phases defined by visual cues.
>
> To further validate this, we have conducted experiments based on the reviewer’s comments regarding tasks where transitions are defined by external, non-physical events. Specifically, we designed a set of special tasks in which the target object does not appear at the beginning of the episode, but instead appears at a random time between 0 and 10 timesteps after the task starts. In these tasks, the policy must wait for a visual cue indicating the appearance of the object before starting the action. As shown in Table 1, GAP achieves significantly better performance than the vision-proprioception policy without GAP in these scenarios, demonstrating that GAP can reliably detect phases defined by visual cues through processing proprioceptive information, and effectively reduces the dependence of the vision-proprioception policy on proprioception during motion-transition phases, thereby improving its performance.
>
> **Table 1: Experiments of tasks that waiting for visual cues**
> | Method | bin-picking (Meta-World) | stack (RoboSuite) | pick cube (Real One-Arm) |
> |:------:|:------:|:---------:|:------------------:|
> | **Vision-only** | 49% | 56% | 10% |
> | **Concatenation** | 31% | 37% | 0% |
> | **GAP** | **55%** | **64%** | **15%** |
>
> $\color{blue}Question \space 2: $ The direct evidence for phase estimation.
>
> $\color{darkgreen}Response \space 2: $
> We appreciate the reviewer’s insightful comment and have included additional visualization in the revised version of our paper. Figure 5 displays the estimated transition probability $\rho$ alongside corresponding RGB images of the task. Additionally, we also show the relationship between $\rho$ and visual uncertainty. We observe that $\rho$ rises sharply during key moments, such as when the robot arm grasps an object or shifts from a vertical movement to a rotational action. This indicates that our approach effectively detects motion transitions in the robot’s operation. Notably, we also find that increases in $\rho$ are consistently accompanied by decreases in visual uncertainty, demonstrating a clear inverse relationship between the two. This suggests that the changes in visual input are relatively subtle during motion transitions, making it important to enhance learning of the visual modality in these moments. These observations provide strong support for the validity of our phase estimation.

---

> ### Author Response · Authors · 2025-11-23
>
> $\color{blue}Question \space 3: $ Comparison plots of the loss curves.
>
> $\color{darkgreen}Response \space 3: $
> We thank the reviewer for this suggestion, as it helps us clearly demonstrate the effectiveness of GAP. We have included the loss curves in Section 6.2 of the revised submission of our work. Due to the application of GAP, the gradient updates for proprioceptive parameters are regulated, which results in a slower loss decrease during the early and middle stages of training. However, the policy will converges to a lower final loss.
>
> $\color{blue}Question \space 4: $ Plot of showing how the phase changes.
>
> $\color{darkgreen}Response \space 4: $
> The corresponding plot has been added as Figure 5 in the revised version of our paper.
>
> $\color{blue}Question \space 5: $ The regulated policy still shows slightly lower performance than the vision-only policy in Figure 4.
>
> $\color{darkgreen}Response \space 5: $
> Figure 4 illustrates an intervention experiment conducted under the same setting as the right panel of Figure 1. In this experiment, we perform the "assembly" task using the vision-only policy, but during a specific 10-timestep interval, we replace the executed actions with those predicted by the vision-proprioception policy under the same observations. Therefore, Figure 4 does not provide a direct performance comparison between the regulated policy and the vision-only policy, as the final task success rate may be affected by the abrupt policy switching during the intervention. However, compared to the vision-proprioception policy without GAP, the regulated policy shows notably better performance during the motion-transition phase, which suggests that GAP effectively enhances the learning of the visual modality during these periods.
>
> $\color{blue}Question \space 6: $ The success rate of the FiLM in Table 4.
>
> $\color{darkgreen}Response \space 6: $
> We suppose that FiLM applies affine transformations to conditionally adjust feature representations, making it particularly well-suited for tasks that require effective modality collaboration. For example, it achieves high success rate on the "push-wall" task, which benefits from such calibration. However, in simpler tasks like "disassemble" where complex conditional adaptation may be unnecessary, FiLM’s performance will decline.

---

> ### Author Response · Authors · 2025-11-23
>
> $\color{blue}Question \space 7: $ The details of bimanual task experiments.
>
> $\color{darkgreen}Response \space 7: $
> We trained separate policies for the left and right arms because the proprioceptive information for each arm only represents its own position, orientation, and gripper opening degree. Furthermore, in bimanual cooperative tasks, they exhibit distinct motion transition phases, making it difficult for a single unified policy to accurately represent motion or estimate transition phases for both arms simultaneously. Therefore, we opted to train independent policies for each arm. We appreciate the reviewer for pointing out this issue and have added explanation in Section A.2 of the revised version of our paper. In future work, modeling not only the individual motions of each arm but also their coordination could enable more expressive representations and phase estimation for bimanual tasks.
>
> $\color{blue}Question \space 8: $ The number of stages for applying GAP.
>
> $\color{darkgreen}Response \space 8: $
> In all our experiments, policies were trained for 100 epochs. We applied GAP during the first 50 epochs to avoid excessive modulation, which may cause unstable gradient updates and potentially lead to policy collapse. To better verify this, we conducted experiments to illustrate how the number of epochs $x$ during which GAP is applied affects the final policy performance. Table 2 shows that applying GAP in the early training stages significantly improves the performance of vision-proprioception policies. However, suppressing the proprioceptive gradients for too many epochs leads to reduced policy performance.
>
> **Table 2: Ablation study on the number of stages for applying GAP $x$**
> | Method | assembly (Meta-World) | threading (RoboSuite) | put the banana in the box (Real One-Arm) |
> |:------:|:------:|:---------:|:------------------:|
> | **Vision-only** | 83% | 44% | 55% |
> | **Concatenation** | 75% | 33% | 50% |
> | **$x=10$** | 77% | 38% | 50% |
> | **$x=30$** | 88% | 47% | 60% |
> | **$x=70$** | 83% | 45% | 55% |
> | **$x=90$** | 72% | 35% | 50% |
> | **GAP ($x=50$)** | **94%** | **53%** | **65%** |
>
> We thank the reviewer for raising these important questions, which help us improve the clarity and completeness of our work. We have added this experiment to Section B.3 of the revised version of our paper.
>
> $\color{blue}Question \space 9: $ The source code of GAP.
>
> $\color{darkgreen}Response \space 9: $
> We have included the source code in the supplementary material to enhance the paper’s replicability and transparency.

---

> ### Author Response · Authors · 2025-11-27
>
> Dear reviewer, we would appreciate knowing if our responses have fully addressed your concerns. We are happy to answer any further questions or comments you may have.

---

> > ### Comment · Reviewer_rSWj · 2025-11-28
> >
> > The rebuttal and the revised paper addressed all my concerns and suggestions. I am happy to raise the score.

---

> > > ### Author Response · Authors · 2025-11-28
> > >
> > > Thank you for constructive suggestions and recognition.

---

### Official Review · Reviewer_X7TA · 2025-10-30

**Soundness:** 3
**Presentation:** 2
**Contribution:** 3
**Rating:** 6
**Confidence:** 5

**Summary:**

This paper proposes GAP, a gradient adjustment algorithm that uses phase-guidance to dynamically modulate proprioception's gradient magnitude during training. This paper identifies a critical issue in vision-proprioception policies: the suppression of visual modality learning during motion-transition phases due to the dominance of more concise proprioceptive signals in optimization. Comprehensive experiments in simulation and real-world settings, across single-arm, dual-arm, and VLA models, demonstrate that GAP significantly improves policy robustness and generalization.

**Strengths:**

1. A detailed analysis was conducted to investigate the reasons behind the suppressed learning of the visual modality;
2. Phase-guided gradient adjustment offers a principled approach to dynamic modality balancing;
3. Comprehensive evaluation across different environments, platforms, and models.

**Weaknesses:**

The assessment lacks a direct evaluation of whether the visual modality is better utilized. While performance metrics provide some evidence, incorporating qualitative visualizations or explainable analysis of the GAP-trained model would more effectively demonstrate the method's effectiveness to readers.

**Questions:**

1. Are the analysis and methods presented in this paper still effective when applied to more advanced policy architectures, such as generalist VLA models built upon imitation learning frameworks with multimodal large language models as their backbone?
2. The paper also includes experiments with a transformer-based VLA (Octo). How did the authors determine parameter attribution and differentiate between parameters belonging to the vision chunk versus the proprioception chunk? Given that Octo's architecture is highly integrated, requiring deep fusion of both observation modalities within the transformer, this distinction seems non-trivial.

---

> ### Author Response · Authors · 2025-11-23
>
> Thank the reviewer for thoughtful comments and insightful suggestions regarding our work. We would like to answer each questions in detail below.
>
> $\color{blue}Question \space 1: $ The direct evaluations of the visual modality.
>
> $\color{darkgreen}Response \space 1: $
> We appreciate the reviewer for raising this important issue, which is crucial for improving the quality of our paper. To directly evaluate the visual modality, we conducted linear-probing experiments on the vision features before and after applying GAP. Specifically, we extract and freeze the visual branch of the vision-proprioception policy and train a separate policy head to predict actions. These policies were evaluated in both in-distribution and out-of-distribution settings (where object positions at test time differ from those in the training data). As shown in Table 1, the visual branch trained with GAP significantly outperforms that without GAP in both in-distribution and out-of-distribution scenarios. This demonstrates that GAP enhances the learning of the visual modality. We have added this experiment to Section C.1 of the revised version of our paper.
>
> **Table 1: The direct evaluations of visual branch**
> | Method | assembly | assembly (OOD) | threading | threading (OOD) |
> |:------:|:------:|:------:|:---------:|:---------:|
> | **Visual branch without GAP** | 61% | 45% | 26% | 14% |
> | **Visual branch with GAP** | **74%** | **69%** | **41%** | **27%** |
>
> $\color{blue}Question \space 2: $ The extension of GAP to VLAs.
>
> $\color{darkgreen}Response \space 2: $
> Investigating how to enhance the collaboration between different modalities within VLAs is an important research direction. As shown in Section 5.4, we conducted an exploration using the Octo model. Furthermore, we conduct additional experiments on OpenVLA-OFT [1] to assess whether GAP remains effective for VLAs. Specifically, we fine-tuned OpenVLA using different input modalities. The results in Table 2 indicate that fine-tuning with both proprioception and vision leads to better performance on VLA tasks compared to using vision only. We hypothesize this is because the visual representations in multimodal large language models are already of high quality, and thus are less subject to suppression during fine-tuning. After applying GAP for adjustment, the performance further improves, suggesting that GAP remains effective for VLAs. Nevertheless, achieving optimal modality collaboration in VLAs warrants further investigation in the future.
>
> **Table 2: Performances of VLA finetuned with different inputs**
> | fine-tuning modality | assembly | bin-picking | threading |
> |:------:|:--------:|:---------:|:-------------------:|
> | **With vision** | 91% | 71% | 59% |
> | **With vision+proprio** | 95% | 77% | 62% |
> | **With vision+proprio (GAP)** | **98%** | **85%** | **77%** |
>
>
> $\color{blue}Question \space 3: $ The details of experiments with transformer-based VLAs.
>
> $\color{darkgreen}Response \space 3: $
> As the reviewer correctly pointed out, in transformer-based models, modalities become highly integrated within the transformer. Therefore, we only applied gradient adjustment to the proprioception feature extractor (i.e., the parameters before the transformer) during training. We appreciate the reviewer’s helpful feedback and have incorporated a detailed clarification of this in Section A.2 of the revised version of our paper.
>
> ---
>
> [1] Kim, Moo Jin, Chelsea Finn, and Percy Liang. "Fine-tuning vision-language-action models: Optimizing speed and success." arXiv preprint arXiv:2502.19645 (2025).

---

> ### Author Response · Authors · 2025-11-27
>
> Dear Reviewer, we truly value your insightful comments. We would appreciate it if you could let us know whether our rebuttal substantially resolves your concerns, or if further discussion would be beneficial.

---

> > ### Comment · Reviewer_X7TA · 2025-11-27
> >
> > Thanks for the updated results. The experimental results have addressed my concerns, and I will increase the score to 8.

---

> > > ### Author Response · Authors · 2025-11-28
> > >
> > > Thank you for the thoughtful comments and valuable suggestions.

---

### Official Review · Reviewer_eFq6 · 2025-10-31

**Soundness:** 3
**Presentation:** 3
**Contribution:** 3
**Rating:** 6
**Confidence:** 2

**Summary:**

The paper argues that in vision+proprioception policies trained by BC, proprio signals dominate optimization (especially at motion-transition phases where vision should matter for target localization) so the visual pathway under-trains and generalization suffers. They propose GAP, which (i) detects motion-consistent phases via CPD on proprio traces, refines transition probabilities with an LSTM, and (ii) down-scales the proprio branch’s gradients during transitions to re-balance learning. GAP improves success rates across Meta-World, RoboSuite, and several real tasks; it also helps a VLA (Octo) variant.

**Strengths:**

- The key idea is that during optimization, proprio-encoder parameters are updated as $\omega_s^{j+1} \leftarrow \omega_s^j - \lambda (1 - \rho)\eta \nabla_{\omega_s^j} \mathcal{L}_{\text{BC}}$, where $\lambda$ controls scaling, $\eta$ is the learning rate, and $\rho \in [0,1]$ measures transition likelihood. A higher $\rho$ (likely transition) down-scales proprio gradients, forcing the visual encoder to learn those steps more effectively. GAP improves success rates across simulated and real tasks (Meta-World, RoboSuite) and enhances vision–proprio fusion even in pretrained VLAs such as Octo. The approach is simple yet effective, but relies on proprio-derived phase signals that might misidentify transitions and suppress useful proprio cues.
- The phase-guided reweighting is simple yet principled. It directly addresses the modality competition problem well known in multimodal learning. The LSTM refinement of $\rho_t$ mitigates the discretization artifacts introduced by CPD.  Together, these are reasonable, low-friction additions to standard behavioral cloning pipelines.
- Key findings of this work include (i) vision-proprio policies can underperform vision-only due to optimization bias toward concise proprio signals; GAP reverses this by reweighting gradients at the right time; (ii) transition phases are the crux: improvements concentrate where target localization is required. Intervention plots show reduced degradation after GAP.

**Weaknesses:**

- Hyperparameters ($\alpha$, $\beta$, $\lambda$) and LSTM size require task-specific tuning, which is a limitation of this work.
- From my understanding, CPD/LSTM finds proprio change points, not visual evidence of new targets. Showing that $\rho_t$ tracks visual uncertainty (e.g., entropy over detectors) would strengthen the claim.
- One minor concern is that the current phase-detection method and the gradient-adjustment strategy both rely on proprioceptive signals (joint positions, velocities, gripper states). Because the change-point detector (CPD) and the transition probability $\rho_t$​ are derived entirely from proprio data, the system may inadvertently bake in proprioceptive priors (already dominate the optimization process). This creates a potential feedback loop: proprio cues define where "transitions" occur, and then proprio gradients are scaled according to those same cues, reinforcing proprio's influence instead of balancing it.

**Questions:**

### Q1: Does GAP play well with on-policy updates (SAC/BCQ) where exploration changes state visitation?

**Action:** Maybe add a small on-policy appendix experiment, and a disturbance test (pushes/perturbations) to ensure GAP doesn’t over-dampen necessary proprio reliance.

### Q2: Is Octo-VP better because the visual branch actually learned more useful features, or simply because the policy became less overfit to proprioception?

**Action:** Linear-probe vision features pre/post GAP and report OOD splits where only visual scene factors change.

### Q3: Does GAP transfer across arms/cameras?

**Action:** Train on robot A, test on robot B (and different wrist cameras) with no re-tuning. This is hard so not required.

---

> ### Author Response · Authors · 2025-11-23
>
> We are truly grateful for the reviewer's positive evaluation of our work and valuable feedback. We take your concerns seriously and address each in detail below.
>
> $\color{blue}Question \space 1: $
> The absence of ablation studies on hyperparameters.
>
> $\color{darkgreen}Response \space 1: $
> We appreciate the reviewer’s concern regarding the absence of ablation studies on hyperparameters, as this raises questions about the need for task-specific tuning. In response, we have conducted additional ablation studies on both simulated environments and real-world to strengthen our paper.
>
> We first evaluate the influence of the hyperparameter $\lambda$ in Equation 5 of our GAP framework, with a default value of $\lambda=0.3$ to control the degree of gradient adjustment. As shown in Table 1, the performance is relatively robust when $\lambda$ varies between 0.2 and 0.4, indicating that precise tuning is not necessary. However, if $\lambda$ is set too low, the reduction of proprioceptive gradients is insufficient, causing the overall performance to approach or even fall below that of the concatenation baseline. On the other hand, setting $\lambda$ too high leads to unstable training and may cause the policy to collapse.
>
> **Table 1: Ablation study on $\lambda$**
> | Method | assembly (Meta-World) | threading (RoboSuite) | put the banana in the box (Real One-Arm) |
> |:------:|:------:|:---------:|:------------------:|
> | **Vision-only** | 83% | 44% | 55% |
> | **Concatenation** | 75% | 33% | 50% |
> | **$\lambda=0.1$** | 77% | 41% | 50% |
> | **$\lambda=0.2$** | 91% | **57%** | 60% |
> | **$\lambda=0.4$** | **96%** | 49% | **65%** |
> | **$\lambda=0.8$** | 70% | 37% | 50% |
> | **GAP ($\lambda=0.3$)** | 94% | 53% | **65%** |
>
> We also performed ablation experiments on the hyperparameters $\alpha$ and $\beta$, which are key to balancing the influence of the different components in our motion-consistency distance (see Equation 4). Specifically, $\alpha$ controls the relative weight between displacement and rotation consistency, while $\beta$ determines the impact of the gripper opening term. In our setup, we assign $\alpha=1$ to treat displacement and rotation with equal importance, as detailed in Section A.2 of the supplementary. Meanwhile, because the gripper opening consistency is measured by a sgn function (yielding 0 or 1), we choose $\beta=2e^{-3}$ so that this contribution remains moderate relative to the other terms. This configuration ensures that the three aspects of motion are appropriately balanced within the metric, and does not allow any single term to dominate.
>
> From Table 2, we observe that tuning $\alpha$ affects the relative emphasis on displacement versus rotation consistency within our distance metric. A lower $\alpha$ favors displacement consistency and thus enhances performance on tasks like "assembly" and "put banana in the box," where positional accuracy is critical. Conversely, a larger $\alpha$ (e.g., $\alpha=2$) prioritizes rotational consistency, which benefits the "threading" task that relies more on precise rotation. Adjusting $\beta$ to a larger value ($10e^{-3}$) does not lead to notable gains across tasks. Importantly, regardless of the settings for $\alpha$ and $\beta$, vision-proprioception policies trained with GAP consistently surpass both the simple concatenation baseline and vision-only policies. This robustness demonstrates that the effectiveness of GAP does not depend on hyperparameter tuning.
>
> **Table 2: Ablation studies on $\alpha$ and $\beta$**
> | Method | assembly | threading | put the banana in the box |
> |:------:|:------:|:---------:|:------------------:|
> | **Vision-only** | 83% | 44% | 55% |
> | **Concatenation** | 75% | 33% | 50% |
> | **$\alpha=0.5, \beta=2e^{-3}$** | 93% | 49% | **65%** |
> | **$\alpha=2, \beta=2e^{-3}$** | 91% | **57%** | 60% |
> | **$\alpha=1, \beta=10e^{-3}$** | 88% | 51% | 60% |
> | **GAP** | **94%** | 53% | **65%** |
>
> For the LSTM, its input size is determined by the dimension of the proprioception information, and for a single embodiment, it does not require tuning. The results in Table 3 show that the size of the LSTM's hidden layer has minimal impact on the final performance.
>
> **Table 3: Ablation study on the hidden layer of LSTM**
> | Method | assembly | threading | put the banana in the box |
> |:------:|:------:|:---------:|:------------------:|
> | **Vision-only** | 83% | 44% | 55% |
> | **Concatenation** | 75% | 33% | 50% |
> | **$hidden\space dim = 128$** | 93% | **57%** | 60% |
> | **$hidden\space dim = 512$** | 91% | 49% | **65%** |
> | **GAP ($hidden\space dim=256$)** | **94%** | 53% | **65%** |
>
> We appreciate the reviewer for raising this important concern, which has helped us improve the quality of our work. We have added the corresponding ablation experiments in Section B.3 of the revised submission.

---

> ### Author Response · Authors · 2025-11-23
>
> $\color{blue}Question \space 2: $ The relationship between $\rho$ and visual uncertainty.
>
> $\color{darkgreen}Response \space 2: $
> During motion transition moments in manipulation tasks, policies need to rely more on visual information for decision-making. However, the visual cues at these moments tends to be subtle. Therefore, we identify change points based on proprioception and enhance the vision modality by adjusting proprioception gradients at these moments during policy training.
>
> We thank the reviewer for this feedback. We have revised the submission and added plots in Section 6.1 to show the estimated transition probabilities $\rho$ along with scene RGB images for validation. Furthermore, we use R3M [1], a vision model specialized for robotics, to extract visual features of the task, and compute local entropy using sliding window covariance as a measure of visual uncertainty $u$ for the task. It is then scaled to the range [0, 1] using $(tanh(u)+1)/2$.
>
> As Figure 6 of the revised submission shown, during motion-consistent phases defined by $\rho$, the R3M-based visual uncertainty is higher, while the scaled visual uncertainty is lower during motion-transition phase. This indicates that visual changes during the motion-transition phase are subtle, thus requiring enhanced learning of the vision modality. Our $\rho$ can effectively distinguish these phases.
>
> Additionally, we substitute the scaled visual uncertainty for $(1-\rho)$ in Equation 5 to investigate whether visual uncertainty can effectively guide gradient adjustment. As shown in Table 4, leveraging visual uncertainty for gradient adjustment also yields noticeable improvements, validating the effectiveness of our gradient adjustment. Because GAP calculates $\rho$ using only CPD and LSTM, it is more computationally efficient than approaches relying on large vision models.
>
> **Table 4: Performances of policies trained with visual uncertainty**
> | Method | assembly | threading | put the banana in the box |
> |:------:|:------:|:---------:|:------------------:|
> | **Vision-only** | 83% | 44% | 55% |
> | **Concatenation** | 75% | 33% | 50% |
> | **Visual uncertainty** | 89% | 51% | **65%** |
> | **GAP** | **94%** | 53% | **65%** |
>
> $\color{blue}Question \space 3: $ The feedback loop of proprioception.
>
> $\color{darkgreen}Response \space 3: $
> Thank you for this insightful question. As detailed in Algorithm 1, GAP comprises two main steps. The first component leverages CPD and LSTM to process proprioceptive information and compute the transition probability $\rho$. The second uses $\rho$ to adjust gradient updates during policy training. Specifically, as described in Equation 5, the gradient updates to proprioceptive parameters are attenuated during motion-transition phases, ensuring that proprioception’s influence on the policy remains balanced rather than being overly emphasized.
>
> $\color{blue}Question \space 4: $ The extension of GAP to on-policy updates.
>
> $\color{darkgreen}Response \space 4: $
> To verify whether GAP can be applied to on-policy updates, we use vision-proprioception policies initialized with behavior cloning and train the LSTM to predict motion transition probabilities in real-time during exploration, thereby adjusting proprioception gradient updates in online learning.
>
> **Table 5: Experiments with on-policy updates**
> | Method | assembly (Meta-World) | bin-picking (Meta-World)| threading (RoboSuite) |
> |:------:|:------:|:---------:|:------------------:|
> | **Vision-only** | 88% | 71% | 50% |
> | **Concatenation** | 78% | 50% | 37% |
> | **GAP** | **94%** | **82%** | **61%** |
>
> The results in Table 5 show that policies trained with online learning are more robust, and GAP can be effectively applied to on-policy updates. Additionally, the online learning experiments demonstrate that GAP does not over-dampen necessary proprioception reliance. We thank the reviewer for raising this valuable suggestion and haved added this experiment to Section C.4 of the revised version of our paper.
>
>
> ---
>
> [1] Nair, Suraj, et al. "R3M: A Universal Visual Representation for Robot Manipulation." Conference on Robot Learning. PMLR, 2023.

---

> ### Author Response · Authors · 2025-11-23
>
> $\color{blue}Question \space 5: $ The evaluation of visual features in OOD scenarios.
>
> $\color{darkgreen}Response \space 5: $
> To verify whether GAP enables the visual branch to actually learn more useful features, or
> simply because the policy became less overfit to proprioception, we conducted linear-probing experiments on the vision features before and after applying GAP. Specifically, we extract and freeze the visual branch of Octo-VP, and then train a separate policy head to predict actions. These policies are evaluated in out-of-distribution (OOD) scenarios (where only the visual scene factors are changed) to assess the impact of applying GAP. The results of Table 6 demonstrate the visual modality trained with GAP significantly outperforms that without GAP. This indicates that GAP enables the visual branch to learn more useful features by slowing down the optimization of proprioception.
>
> **Table 6: The evaluation of visual features in OOD scenarios**
> | Method | disassemble (Meta-World) | push-wall (Meta-World)| threading (RoboSuite) |
> |:------:|:------:|:---------:|:------------------:|
> | **Visual branch of Octo-VP** | 66% | 37% | 43% |
> | **Visual branch of Octo-VP with GAP** | **75%** | **48%** | **56%** |
>
> $\color{blue}Question \space 6: $ Does GAP transfer across arms/cameras?
>
> $\color{darkgreen}Response \space 6: $
> We thank the reviewer for raising this interesting question. Transferring across different robotic arms presents significant challenges for models other than Vision-Language-Action (VLA) models, as the semantics of proprioceptive information vary between arms. This makes it difficult to apply GAP directly to a new arm without additional adaptation or re-tuning. Nevertheless, we investigate the transferability of GAP across camera perspectives by altering the camera position during testing. In particular, we shift the camera 3cm to the right and evaluate the policy’s performance in this new setting without any further re-tuning.
>
> The results in Table 7 show that although vision-only and vision-proprioception policies without GAP perform poorly in this setting, the maintained superior performances indicate the effectiveness of GAP.
>
> **Table 7: The evaluation of transfer across cameras**
> | Method | assembly | bin-picking | threading |
> |:------:|:------:|:---------:|:------------------:|
> | **Vision-only** | 71% | 57% | 38% |
> | **Concatenation** | 66% | 42% | 29% |
> | **GAP** | **79%** | **64%** | **44%** |

---

> > ### Comment · Reviewer_eFq6 · 2025-11-26
> >
> > Thanks for the update. I raised my confidence of the evaluation.

---

> > > ### Author Response · Authors · 2025-11-26
> > >
> > > Thank you for the helpful suggestions and recognition.

---

### Official Review · Reviewer_mBpw · 2025-11-02

**Soundness:** 4
**Presentation:** 4
**Contribution:** 3
**Rating:** 8
**Confidence:** 3

**Summary:**

This paper explores an efficient strategy for combining vision observations with proprioception observations in robotic manipulation. The authors first empirically show that naïve vision-proprioception policies tend to underutilize vision during motion-transition phases, which degrades performance compared to pure vision policies. To address this, they propose a Gradient Adjustment with Phase-guidance (GAP) method that dynamically scales the proprioceptive gradient updates during training, particularly for parameters related to motion-transition estimation. Experimental results demonstrate that GAP effectively improves both in-distribution and out-of-distribution (OOD) performance.

**Strengths:**

1. The paper tackles an important and practical challenge in robot learning—how to effectively combine vision and proprioception for efficient and accurate manipulation.
2. The empirical findings are well-motivated, and the proposed gradient adjustment method, GAP, is simple, interpretable, and effective.
3.  Both simulation and real-world experiments demonstrate the effectiveness of GAP. The approach shows consistent gains over both vision-only and vision-proprioception baselines, and the improved OOD performance is particularly promising.

**Weaknesses:**

Overall, this is a good paper for me, but there are a few minor concerns:
1. The paper lacks a deeper investigation into the form of the gradient adjustment function $(1-p)$ and the choice of key hyperparameters. Since the paper mentions that gradient adjustment is applied only during the early stage of training, it would be beneficial to discuss the rationale for choosing specific hyperparameters ($\alpha $, $\beta$, and the number of stages for applying GAP) and to include an ablation study on parameter sensitivity.
2. GAP adjusts gradients for certain parameters of the network. This may have connections to GradNorm-based balancing methods [1]. For instance, if early in training the gradient magnitude of the proprioceptive branch is larger than that of the vision branch, then GAP might behave similarly to gradient normalization. It would be helpful to clarify whether GAP implicitly aligns with GradNorm principles. Including statistics of gradient norms for both branches and showing how the adjustment parameter $p$ evolves during training would strengthen the analysis.

[1] Chen, Zhao, et al. “GradNorm: Gradient Normalization for Adaptive Loss Balancing in Deep Multitask Networks.” ICML 2018.

**Questions:**

1) What is the reasoning behind the choice of GAP hyperparameters ($\alpha $, $\beta$, and stage schedule)? How sensitive is the method to these values?
2) What is the relationship between GAP and GradNorm-based gradient balancing? Would a GradNorm-style adjustment (e.g., scaling by grad/∥grad∥) yield similar improvements?

---

> ### Author Response · Authors · 2025-11-23
>
> We sincerely appreciate the reviewer's recognition of our work and the valuable feedback. We have carefully addressed each of your concerns and questions in detail below.
>
> $\color{blue}Question \space 1: $
> The ablation studies on gradient adjustment function and hyperparameters.
>
> $\color{darkgreen}Response \space 1: $
> We thank the reviewer for this valuable feedback. To address this, we have provided detailed descriptions and supplemented ablation studies on both simulated environments and real-world in Section B.3 of the revised version of our paper.
>
> **The form of the gradient adjustment function $(1-\rho)$:** In this work, we applied Equation 5 in our paper to adjust the gradient updating of the proprioception modality:
>
> $$
> \omega_s^{j+1} = \omega_s^j - \lambda \cdot (1-\rho) \cdot \eta \nabla_{\omega_s^j} \mathcal{L}_{BC}(\omega_s^j)
> $$
>
> To ensure that a higher value of the motion-transition phase indicator $\rho$ results in slower gradient updates, we use the form $(1-\rho)$ in combination with $\lambda$ to adjust the gradient. This choice is both natural and appropriate, given that $\rho$ takes values in the range $[0, 1]$. To address the reviewer's concerns, we keep hyperparameters such as $\lambda$ fixed and compare $(1-\rho)$ with the following alternative adjustment functions:
>
> - **Sin**: We use $(1-\text{sin}(\rho))$ to control the magnitude of gradient adjustment. After mapping through the sin function, the overall adjustment strength becomes smaller but smoother.
>
> - **Tanh**: Similar to the sin function, the gradient adjustment function $(1-\text{tanh}(\rho))$ can also make the adjustment smoother.
>
> - **Sigmoid**: We use the form $(1-\text{sigmoid}(8(\rho-0.5)))$ so that for input $\rho$, its output range is approximately within [0-1].
>
> As shown in Table 1, different gradient adjustment functions can mitigate the suppression of the vision modality by modulating the gradient updates of the proprioception modality. Among them, the sin and tanh variants result in slightly lower performance improvements, likely due to their reduced adjustment strength. The scaled sigmoid function, on the other hand, becomes excessively steep, which can lead to numerical instability during training. In contrast, the $(1-\rho)$ adjustment function that we adopt is straightforward, and achieves consistently better results.
>
> **Table 1: Ablation study on the gradient adjustment function**
> | Method | assembly (Meta-World) | threading (RoboSuite) | put the banana in the box (Real One-Arm) |
> |:------:|:------:|:---------:|:------------------:|
> | **Vision-only** | 83% | 44% | 55% |
> | **Concatenation** | 75% | 33% | 50% |
> | **Sin** | 88% | 52% | 60% |
> | **Tanh** | 90% | 50% | 60% |
> | **Sigmoid** | 85% | 47% | 55% |
> | **GAP (Ours)** | **94%** | **53%** | **65%** |
>
> Furthermore, we also conducted an ablation study on the hyperparameter $\lambda$ in Equation 5. In our GAP framework, we set $\lambda=0.3$ by default to control the strength of gradient adjustment. As shown in Table 2, the policy performance is not highly sensitive to changes in $\lambda$ between 0.2 and 0.4, and does not require precise fine-tuning. However, if $\lambda$ is set too small, the gradients of the proprioceptive modality are not sufficiently attenuated, causing the policy’s performance to approach (or even drop below) that of the simple concatenation baseline. Conversely, if $\lambda$ is set too large, it can lead to instability or even collapse during policy training.
>
> **Table 2: Ablation study on $\lambda$**
> | Method | assembly | threading | put the banana in the box |
> |:------:|:------:|:---------:|:------------------:|
> | **Vision-only** | 83% | 44% | 55% |
> | **Concatenation** | 75% | 33% | 50% |
> | **$\lambda=0.1$** | 77% | 41% | 50% |
> | **$\lambda=0.2$** | 91% | **57%** | 60% |
> | **$\lambda=0.4$** | **96%** | 49% | **65%** |
> | **$\lambda=0.8$** | 70% | 37% | 50% |
> | **GAP ($\lambda=0.3$)** | 94% | 53% | **65%** |

---

> ### Author Response · Authors · 2025-11-23
>
> **Hyperparameters of $\alpha, \beta$ and the number of stages for applying GAP $x$:**
>
> The selection of $\alpha$ and $\beta$ serves to balance the relative contribution of each term in Equation 4, thus providing a more accurate measurement of whether the directions of these changes are consistent. As described in Section A.2 of the supplementary material, we set $\alpha$ to 1 so that displacement consistency and rotation consistency are equally weighted. Since gripper opening consistency is measured via the sign function (which outputs 0 or 1), we set $\beta=2 e^{-3}$ to ensure this term won't overshadow the first two terms. In addition, we also performed ablation studies on hyperparameters $\alpha$, $\beta$, and the number of training stages, denoted as $x$, during which GAP is applied.
>
> **Table 3: Ablation studies on $\alpha$ and $\beta$**
> | Method | assembly | threading | put the banana in the box |
> |:------:|:----:|:------:|:--------:|
> | **Vision-only** | 83% | 44% | 55% |
> | **Concatenation** | 75% | 33% | 50% |
> | **$\alpha=0.5, \beta=2e^{-3}$** | 93% | 49% | **65%** |
> | **$\alpha=2, \beta=2e^{-3}$** | 91% | **57%** | 60% |
> | **$\alpha=1, \beta=10e^{-3}$** | 88% | 51% | 60% |
> | **GAP** | **94%** | 53% | **65%** |
>
> As indicated in Table 3, a smaller value of $\alpha$ increases the influence of displacement consistency on overall motion consistency, resulting in better outcomes for tasks dominated by displacement, such as "assembly" and "put banana in the box." Conversely, setting $\alpha=2$ makes the policy perform best on the "threading" task, where rotation plays a more crucial role. Adjusting $\beta$ up to $10e^{-3}$ in order to highlight the effect of gripper opening does not bring significant gains. Importantly, regardless of the specific hyperparameter configuration, all three choices enable the vision-proprioception policy with GAP to surpass both the concatenation baseline and the vision-only policy. This demonstrates that the effectiveness of our proposed GAP method is robust to hyperparameter selection.
>
> In all of our experiments, policies were trained for 100 epochs. We applied GAP during the first 50 epochs to avoid excessive modulation, which may cause unstable gradient updates and potentially lead to policy collapse. Table 4 presents the impact of the number of training stages $x$ during which GAP is applied on the final policy performance. Applying GAP in the early stages of training does improve the performances of vision-proprioception policies. However, if the gradient updates for proprioception are suppressed for too many epochs, the performance of policies would deteriorate.
>
> **Table 4: Ablation study on the number of stages for applying GAP $x$**
> | Method | assembly | threading | put the banana in the box |
> |:------:|:------:|:---------:|:--------:|
> | **Vision-only** | 83% | 44% | 55% |
> | **Concatenation** | 75% | 33% | 50% |
> | **$x=10$** | 77% | 38% | 50% |
> | **$x=30$** | 88% | 47% | 60% |
> | **$x=70$** | 83% | 45% | 55% |
> | **$x=90$** | 72% | 35% | 50% |
> | **GAP ($x=50$)** | **94%** | **53%** | **65%** |
>
>
> $\color{blue}Question \space 2: $
> The relationship between GAP and GradNorm-based gradient balancing.
>
> $\color{darkgreen}Response \space 2: $
> This is a insightful question. Both GradNorm and GAP aim to achieve balanced learning by adjusting gradients, and the forms of their gradient adjustments are similar. The key difference lies in how the magnitude of the adjustment is determined. GAP predicts the phase transition probability based on proprioception and sets the adjustment magnitude according to these probabilities, which do not change across different training epochs. In contrast, GradNorm adaptively adjusts gradients based on the loss ratios of tasks in each epoch within a multi-task learning framework.
>
> Therefore, in the context where proprioception and vision modalities are fused to predict actions, directly applying GradNorm is not suitable for such single-loss training. As an alternative, one can use the vision and proprioception features separately to predict actions within the original policy training framework, considering them as auxiliary tasks. Gradient balancing is then applied according to the loss ratios of these auxiliary tasks. However, as shown in Table 5, this approach does not lead to significant performance improvements. We suppose this is because the proprioceptive modality lacks information about the target object and thus can not predict actions independently, making it unreasonable to use as an auxiliary task. In contrast, GAP adjusts gradients based on phase transitions, which is more suitable for robotic manipulation tasks.
>
> **Table 5: Performance comparison between GAP and GradNorm**
> | Method | assembly | threading | put the banana in the box |
> |:------:|:------:|:---------:|:-------------:|
> | **Vision-only** | 83% | 44% | 55% |
> | **Concatenation** | 75% | 33% | 50% |
> | **GradNorm** | 86% | 35% | 50% |
> | **GAP** | **94%** | **53%** | **65%** |

---

> ### Author Response · Authors · 2025-11-27
>
> Dear reviewer, we would be grateful to receive your feedback regarding whether our responses have satisfactorily addressed the your concerns. Please let us know if there are any remaining issues that would benefit from further discussion.

---

### Author Response · Authors · 2025-11-30
**Summary of rebuttal**

Dear Area Chair and Reviewers,

As the rebuttal period draws to a close, we would like to provide a brief summary of our responses.

First, we sincerely thank all reviewers and the Area Chair for their constructive feedback and the time spent evaluating our work. All reviewers recognize the contribution of our analysis of the inconsistent performance of vision-proprioception policies. We are also grateful that the reviewers affirmed the significant performance improvements brought by GAP across different environments, platforms, and policy architectures.

Meanwhile, some shared concerns were raised, and we have addressed them in detail. **After rebuttal all reviewers express positive ratings toward the paper. Our detailed responses and additional experiments have increased Reviewer eFq6's confidence to 4. The direct evaluation of the visual modality led Reviewer X7TA to raise the score to 8. Reviewer rSWj remarked that our rebuttal and revised paper addressed all concerns and suggestions.**

__Ablations on hyperparameter of GAP__

We have conducted additional ablation studies on both simulated environments and real-world settings to address this concern. The robustness of GAP across different hyperparameter settings demonstrates that the effectiveness of GAP does not depend on extensive hyperparameter tuning. We have added the corresponding ablation experiments in Section B.3 of the revised submission.

__Direct evaluation of the visual modality after applying GAP__

We conducted linear-probing experiments on the vision features before and after applying GAP. The visual modality trained with GAP significantly outperforms that without GAP in both in-distribution and out-of-distribution scenarios, demonstrating that GAP enhances the learning of the visual modality. We have added this experiment to Section C.1 of the revised version of our paper.

__Visualization of the GAP mechanism__

In Figure 5 of the revised submission, we display the estimated transition probability $\rho$ alongside corresponding RGB images of the task. Additionally, we show the relationship between $\rho$ and visual uncertainty. As we have detailed in our rebuttal, this figure provides strong support for the validity of our phase estimation.

We have also included the loss curves in Section 6.2. Due to the use of GAP, the gradient updates for proprioceptive parameters are regulated, which results in a slower loss decrease during the early and middle stages of training. However, the policy converges to a lower final loss.

Based on the rebuttal exchanges, we are encouraged that our additional analyses and clarifications have alleviated the reviewers' concerns. The new ablations, evaluations, and visualizations help to significantly improve the quality of our paper.

We once again thank the reviewers and the Area Chair for their time, insightful comments, and constructive guidance.

Best regards,

Submission12240 Authors

---

### Meta-Review · Area_Chair_zjEW · 2025-12-30

**Summary:**

The paper proposes the GAP method, which dynamically adjusts gradients using phase guidance to combine vision and proprioception for robotic manipulation.

The reviewers acknowledge the simplicity and effectiveness of the proposed approach, as well as the strong empirical results. The initial reviewer scores were 8, 6, 6, and 4. The main concerns raised in the initial reviews included:
 (1) the choice and tuning of key hyperparameters,
 (2) the reliance on proprioceptive cues for phase detection, and
 (3) the lack of qualitative visualizations.

The rebuttal adequately addressed these concerns. All reviewers agreed to accept the paper.
Based on the positive reviewer consensus, the AC recommends accepting the paper.

**Reviewer Concerns:**

See above.

**Reviewer Scores:**

Reviewers increased scores.

---

### Decision · Program_Chairs · 2026-01-26

Accept (Poster)